# PROMPT-TO-PROMPT IMAGE EDITING WITH CROSS-ATTENTION CONTROL

Amir Hertz[*1,2], Ron Mokady[*1,2], Jay Tenenbaum [1], Kfir Aberman[1], Yael Pritch[1], and Daniel Cohen-Or[*1,2]

[1] *Google Research*
[2]*The Blavatnik School of Computer Science, Tel Aviv University*

## ABSTRACT

Recent large-scale text-driven synthesis diffusion models have attracted much attention thanks to their remarkable capabilities of generating highly diverse images that follow given text prompts. Therefore, it is only natural to build upon these synthesis models to provide text-driven image editing capabilities. However, Editing is challenging for these generative models, since an innate property of an editing technique is to preserve some content from the original image, while in the text-based models, even a small modification of the text prompt often leads to a completely different outcome. State-of-the-art methods mitigate this by requiring the users to provide a spatial mask to localize the edit, hence, ignoring the original structure and content within the masked region. In this paper, we pursue an intuitive *prompt-to-prompt* editing framework, where the edits are controlled by text only. We analyze a text-conditioned model in depth and observe that the cross-attention layers are the key to controlling the relation between the spatial layout of the image to each word in the prompt. With this observation, we propose to control the attention maps of the edited image by injecting the attention maps of the original image along the diffusion process. Our approach enables us to monitor the synthesis process by editing the textual prompt only, paving the way to a myriad of caption-based editing applications such as localized editing by replacing a word, global editing by adding a specification, and even controlling the extent to which a word is reflected in the image. We present our results over diverse images and prompts with different text-to-image models, demonstrating high-quality synthesis and fidelity to the edited prompts.

## 1 INTRODUCTION

Recently, large-scale language-image (LLI) models, such as Imagen (Saharia et al., 2022b), DALL·E 2 (Ramesh et al., 2022) and Parti (Yu et al., 2022), have shown phenomenal generative semantic and compositional power, and gained unprecedented attention from the research community and the public eye. These LLI models are trained on extremely large language-image datasets and use state-of-the-art image generative models including auto-regressive and diffusion models. However, these models do not provide simple editing means, and generally lack control over specific semantic regions of a given image. In particular, even the slightest change in the textual prompt may lead to a completely different output image. To circumvent this, LLI-based methods (Nichol et al., 2021; Avrahami et al., 2022a; Ramesh et al., 2022) require the user to explicitly mask a part of the image to be inpainted, and drive the edited image to change in the masked area only, while matching the background of the original image. This approach has provided appealing results, however, the masking procedure is cumbersome, hampering quick and intuitive text-driven editing. Moreover, masking the image content removes important structural information, which is completely ignored in the inpainting process. Therefore, some capabilities are out of the inpainting scope, such as modifying the texture of a specific object.

In this paper, we introduce an intuitive and powerful *textual editing* method to semantically edit images in pre-trained text-conditioned diffusion models via *Prompt-to-Prompt* manipulations. To do so, we dive deep into the cross-attention layers and explore their semantic strength as a handle to control the generated image. Specifically, we consider the internal *cross-attention maps*, which are

---

[*]Performed this work while working at Google.

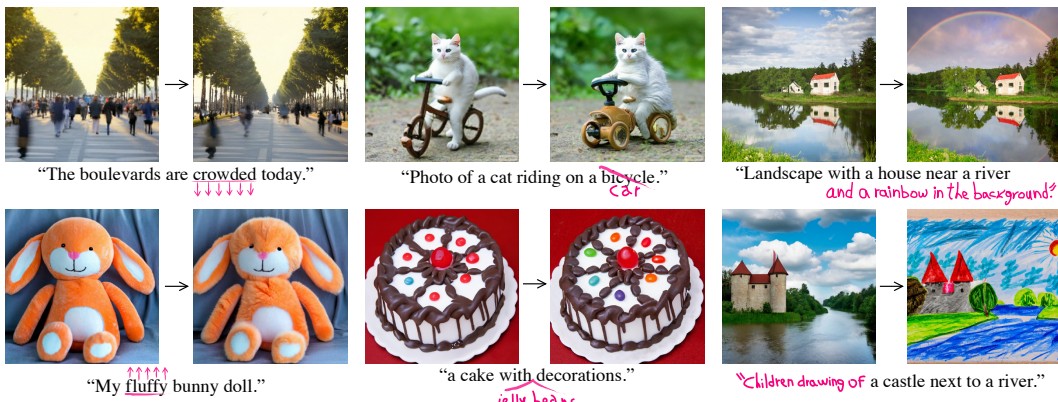

Figure 1: ***Prompt-to-Prompt*** **editing capabilities.** *Our method paves the way for a myriad of caption-based editing operations: tuning the level of influence of an adjective word (bottom-left), making a local modification in the image by replacing or adding a word (bottom-middle), or specifying a global modification (bottom-right).*

high-dimensional tensors that bind pixels and tokens extracted from the prompt text. We find that these maps contain rich semantic relations which critically affect the generated image.

Our key idea is that we can edit images by injecting the cross-attention maps during the diffusion process, controlling which pixels attend to which tokens of the prompt text during which diffusion steps. To apply our approach to various creative editing applications, we show several methods to control the cross-attention maps through a simple and semantic interface (see fig. 1). The first is to change a single token's value in the prompt (e.g., "dog" to "cat"), while fixing the cross-attention maps, to preserve the scene composition. The second is adding new words to the prompt and freezing the attention on previous tokens while allowing new attention to flow to the new tokens. This enables us to perform global editing or modify a specific object. The third is to amplify or attenuate the semantic effect of a word in the generated image. Furthermore, we demonstrate how to use these attention maps to obtain a local editing effect that accurately preserves the background.

Our approach constitutes an intuitive image editing interface through editing only the textual prompt, therefore called *Prompt-to-Prompt*. This method enables various editing tasks, which are challenging otherwise, and does not require model training, fine-tuning, extra data, or optimization. Throughout our analysis, we discover even more control over the generation process, recognizing a trade-off between the fidelity to the edited prompt and the source image. We also demonstrate that our method operates with different text-to-image models as a backbone and we will publish our code for the public models upon acceptance. Finally, our method even applies to real images by using an existing inversion technique. Our experiments show that our method enables intuitive text-based editing over diverse images that current methods struggle with.

## 2 RELATED WORK

Image editing is one of the most fundamental tasks in computer graphics, encompassing the process of modifying an input image through the use of an auxiliary input, such as a label, mask, or reference image. A specifically intuitive way to edit an image is through textual prompts provided by the user. Recently, text-driven image manipulation has achieved significant progress using GANs (Goodfellow et al., 2014; Brock et al., 2018; Karras et al., 2019), which are known for their high-quality generation, in tandem with CLIP (Radford et al., 2021), which consists of a semantically rich joint image-text representation, trained over millions of text-image pairs. Seminal works (Patashnik et al., 2021; Gal et al., 2021; Xia et al., 2021a) which combined these components were revolutionary, since they did not require extra manual labor, and produced realistic manipulations using text only. For instance, Bau et al. (2021) further demonstrated how to use masks to restrict the text-based editing to a specific region. However, while GAN-based editing approaches succeed on curated data, e.g., human faces, they struggle over large and diverse datasets (Mokady et al., 2022).

To obtain more expressive generation capabilities, Crowson et al. (2022) use VQ-GAN (Esser et al., 2021b), trained over diverse data, as a backbone. Other works (Avrahami et al., 2022b; Kim et al., 2022) exploit the recent Diffusion models (Ho et al., 2020; Song & Ermon, 2019; Ho et al., 2020; Song et al., 2020; Rombach et al., 2021; Ho et al., 2022; Saharia et al., 2021; 2022a), which achieve state-of-the-art generation quality over diverse datasets, often surpassing GANs (Dhariwal & Nichol, 2021). Kim et al. (2022) show how to perform global changes, whereas Avrahami et al. (2022b) suc-

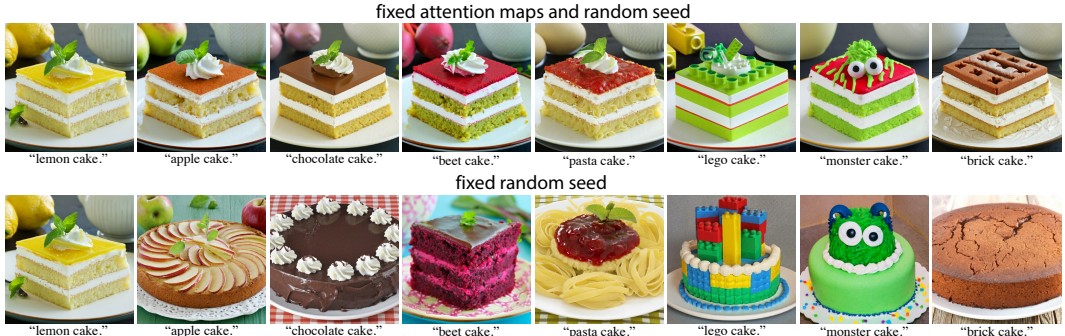

Figure 2: **Content modification through attention injection.** *We start from an original image generated from the prompt "lemon cake" (top left), and modify the text prompt to a variety of other cakes. On the top row, we inject the attention weights of the original image during the diffusion process. On the bottom, we only use the same random seeds as the original image, without injecting attention. The latter leads to a completely new structure that is hardly related to the original.*

cessfully perform local manipulations using user-provided masks for guidance. While most works that require only text (i.e., no masks) are limited to global editing (Crowson et al., 2022; Kwon & Ye, 2021), Bar-Tal et al. (2022) proposed a text-based localized editing technique without using any mask, showing impressive results. Yet, their techniques mainly allow changing textures, but not modifying complex structures, such as changing a bicycle to a car. Moreover, unlike our method, their approach requires training a network for each input.

Numerous works (Ding et al., 2021; Hinz et al., 2020; Tao et al., 2020; Li et al., 2019; Ramesh et al., 2021; Zhang et al., 2018b; Crowson et al., 2022; Gafni et al., 2022; Rombach et al., 2021) advanced the generation of images conditioned on plain text, known as text-to-image synthesis. But only recently these were followed by several large-scale text-image models, such as Imagen (Saharia et al., 2022b), DALL-E2 (Ramesh et al., 2022), and Parti (Yu et al., 2022), demonstrating unprecedented semantic generation. However, these models do not provide control over a generated image, specifically using text guidance only. Changing a single word in the original prompt associated with the image often leads to a completely different outcome. For instance, adding the adjective "white" to "dog" often changes the dog's shape. To overcome this, several works (Nichol et al., 2021; Avrahami et al., 2022a) assume that the user provides a mask to restrict the edited region.

Unlike previous works, our method requires textual input only, by using the spatial information from the internal layers of the generative model itself. This offers the user a much more intuitive editing experience of modifying local or global details by merely modifying the text prompt.

## 3 METHOD

Let $\mathcal{I}$ be an image that was generated by a text-guided diffusion model using the text prompt $\mathcal{P}$ and a random seed $s$. Our goal is to edit $\mathcal{I}$, using only the guidance of an edited prompt $\mathcal{P}^*$, in order to get an edited image $\mathcal{I}^*$ that maintains the content and structure of the original image but corresponds to the edited prompt. For example, consider an image generated from the prompt "my new bicycle", and assume that the user wants to edit the color of the bicycle or replace it with a scooter while preserving the appearance and structure of the original image. An intuitive interface for the user is to directly change the text prompt by further describing the appearance of the bike, or replacing it with another word, respectively. As opposed to previous works, we wish to avoid relying on any user-defined mask to assist or signify where the edit should occur. A simple, but unsuccessful attempt is to fix the internal randomness and regenerate using the edited text prompt. Unfortunately, as fig. 2 shows, this results in a completely different structure and composition.

Our key observation is that the structure and appearance of the generated image depend not only on the random seed, but also on the *interaction* between the pixels to the text embedding through the diffusion process. By modifying the pixel-to-text interaction that occurs in *cross-attention* layers, we provide Prompt-to-Prompt image editing capabilities. More specifically, injecting the cross-attention maps of the input image $\mathcal{I}$ enables us to preserve the original composition and structure. In Section 3.1, we review how cross attention is used, and in Section 3.2, we describe how to exploit the cross-attention for editing. Self-attention is discussed in section 3.3. For background on diffusion models, refer to appendix A.

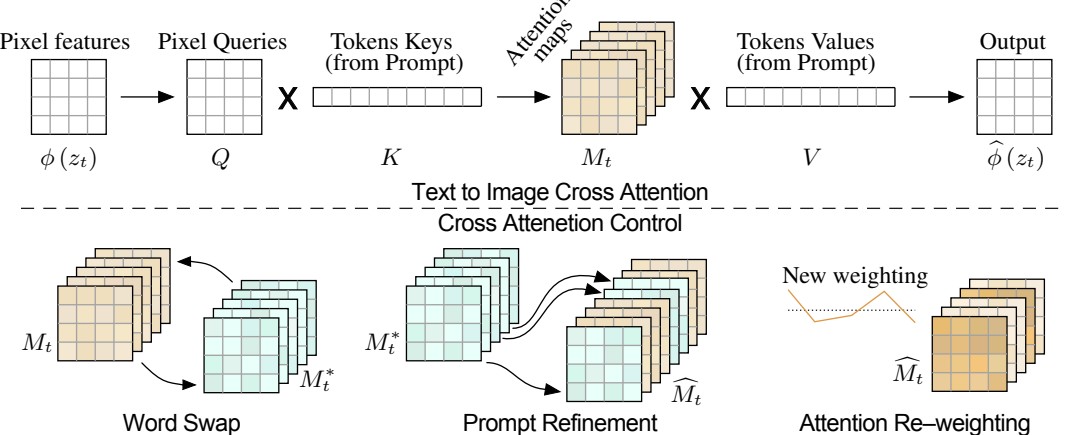

Figure 3: **Method overview.** *Top: visual and textual embedding are fused using cross-attention layers that produce attention maps for each textual token. Bottom: we control the spatial layout and geometry of the generated image using the attention maps of a source image. This enables various editing tasks through editing the textual prompt only. When swapping a word in the prompt, we inject the source image maps $M_t$, overriding the target maps $M_t^*$. In the case of adding a refinement phrase, we inject only the maps that correspond to the unchanged part of the prompt. To amplify or attenuate the semantic effect of a word, we re-weight the corresponding attention map.*

## 3.1 CROSS-ATTENTION IN TEXT-CONDITIONED DIFFUSION MODELS

In this section, we refer to the Imagen (Saharia et al., 2022b) text-guided synthesis model as our backbone, although our method is not limited to a specific model, and results with Latent Diffusion and Stable Diffusion (Rombach et al., 2021) are presented in section 4.1 and appendix C. All three models condition on the text prompt in the noise prediction of each diffusion step throughout cross-attention layers. For further details about the attention layers within each model, please see appendix A.2. Since the composition and geometry are mostly determined at the $64 \times 64$ resolution, we only adapt the text-to-image diffusion model, using the super-resolution process as is. Recall that each diffusion step $t$ consists of predicting the noise $\epsilon$ from a noisy image $z_t$ and text embedding $\psi(\mathcal{P})$ using a U-shaped network (Ronneberger et al., 2015). At the final step, this process yields the generated image $\mathcal{I} = z_0$. Most importantly, the interaction between the two modalities occurs during the noise prediction, where the embeddings of the visual and textual features are fused using cross-attention layers that produce spatial attention maps for each textual token. More formally, as illustrated in fig. 3 (top), the deep spatial features of the noisy image $\phi(z_t)$ are projected to a query matrix $Q = \ell_Q(\phi(z_t))$, and the textual embedding is projected to a key matrix $K = \ell_K(\psi(\mathcal{P}))$ and a value matrix $V = \ell_V(\psi(\mathcal{P}))$, via learned linear projections $\ell_Q, \ell_K, \ell_V$. *Attention maps* are then

$$M = \text{Softmax}\left(\frac{QK^T}{\sqrt{d}}\right), \tag{1}$$

where the cell $M_{ij}$ defines the weight of the value of the $j$-th token on the pixel $i$, and $d$ is the latent projection dimension of the keys and queries. Finally, the cross-attention output is defined to be $\widehat{\phi}(z_t) = MV$, which is then used to update the spatial features $\phi(z_t)$.

Intuitively, the cross-attention output $MV$ is a weighted average of the values $V$ where the weights are the attention maps $M$, which are correlated to the *similarity* between $Q$ and $K$. In practice, to increase their expressiveness, multi-head attention (Vaswani et al., 2017) is used in parallel, and then the results are concatenated and passed through a learned linear layer to get the final output.

## 3.2 CONTROLLING THE CROSS-ATTENTION

We return to our key observation — the spatial layout and geometry of the generated image depend on the *cross-attention* maps. The interaction between pixels and text is illustrated in fig. 4, where the average attention maps are plotted. As can be seen, pixels are more *attracted* to the words that describe them, e.g., pixels of the bear are correlated with the word "bear". Note that averaging is done for visualization purposes, and attention maps are kept separate for each head. Interestingly, we can see that the structure is already determined in the early steps of the diffusion process.

Since the attention reflects the overall composition, we can inject the attention maps $M$ that were obtained from the generation with the original prompt $\mathcal{P}$, into a second generation with the modified prompt $\mathcal{P}^*$. This allows the synthesis of an edited image $\mathcal{I}^*$ that is not only manipulated according

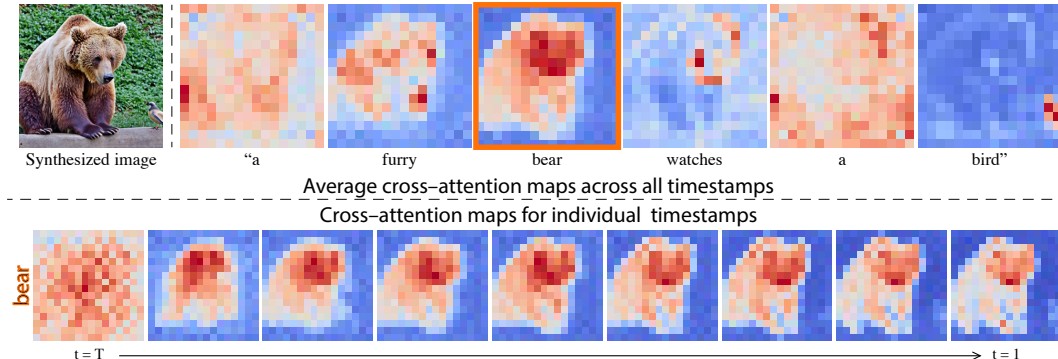

Figure 4: **Cross-attention maps of a text-conditioned diffusion image generation.** *Top: average attention masks for each word in the prompt which was used to synthesize the left image. Bottom: attention maps with respect to the word "bear" from different diffusion steps, ranging from the first step $T = 256$ to the last step $t = 1$ in equal intervals.*

to the edited prompt, but also preserves the structure of the input image $\mathcal{I}$. This is a specific instance of a broader set of attention-based manipulations leading to different types of intuitive editing. We, therefore, start by proposing a general framework, followed by the details of the specific operations.

Let $DM(z_t, \mathcal{P}, t, s)$ be the computation of a single step $t$ of the diffusion process, which outputs the noisy image $z_{t-1}$, and the attention map $M_t$ (omitted if not used). We denote by $DM(z_t, \mathcal{P}, t, s)\{M \leftarrow \widehat{M}\}$ the diffusion step where we override the attention map $M$ with an additional given map $\widehat{M}$, but keep the values $V$ from the supplied prompt. We also denote by $M_t^*$ the produced attention map using the edited prompt $\mathcal{P}^*$. Lastly, we define $Edit(M_t, M_t^*, t)$ to be a general edit function, receiving as input the $t$'th attention maps of the original and edited images.

Our general algorithm for controlled generation consists of performing the iterative diffusion process for both prompts simultaneously, where an attention-based manipulation is applied in each step according to the desired editing task. We fix the internal randomness since even for the same prompt, two random seeds produce drastically different outputs. We also define a local editing scheme in a subsequent paragraph. Formally, our general algorithm for editing the image $\mathcal{I}$, which is generated by prompt $\mathcal{P}$ and seed $s$, is defined:

---

**Algorithm 1:** Prompt-to-Prompt image editing

1   **Input:** A source prompt $\mathcal{P}$, a target prompt $\mathcal{P}^*$, and a random seed $s$.
2   **Optional for local editing:** $w$ and $w^*$, words in $\mathcal{P}$ and $\mathcal{P}^*$, specifying the editing region.
3   **Output:** A source image $x_{src}$ and an edited image $x_{dst}$.
4   $z_T \sim N(0, I)$ a unit Gaussian random variable with random seed $s$;
5   $z_T^* \leftarrow z_T$;
6   **for** $t = T, T-1, \ldots, 1$ **do**
7      $z_{t-1}, M_t \leftarrow DM(z_t, \mathcal{P}, t, s)$;
8      $M_t^* \leftarrow DM(z_t^*, \mathcal{P}^*, t, s)$;
9      $\widehat{M}_t \leftarrow Edit(M_t, M_t^*, t)$;
10      $z_{t-1}^* \leftarrow DM(z_t^*, \mathcal{P}^*, t, s)\{M \leftarrow \widehat{M}_t\}$;
11      **if** *local* **then**
12         $\alpha \leftarrow B(\overline{M}_{t,w}) \cup B(\overline{M}_{t,w^*}^*)$;
13         $z_{t-1}^* \leftarrow (1 - \alpha) \odot z_{t-1} + \alpha \odot z_{t-1}^*$;
14      **end**
15   **end**
16   **Return** $(z_0, z_0^*)$

---

For editing real images, see section 4. Also, note that we can skip the forward call in line 8 by applying the edit function inside the diffusion forward function. Moreover, a diffusion step can be applied on both $z_{t-1}$ and $z_t^*$ in the same batch (i.e., in parallel). We now turn to address local editing followed by specific editing operations, filling the missing definition of the $Edit(M_t, M_t^*, t)$ function. An overview is presented in fig. 3(Bottom).

**Local Editing.** In a common scenario, the user would like to modify a specific object or region, while preserving the rest of the details (i.e., background). For this purpose, we utilize the cross-

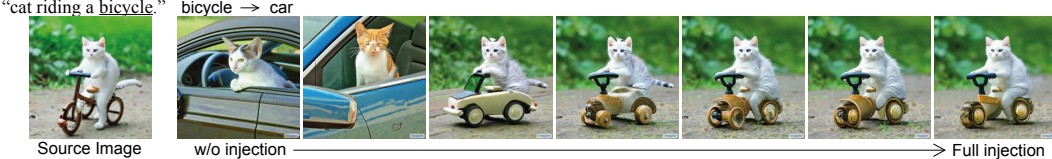

Figure 5: **Attention injection through a varied number of diffusion steps.** *We edit the image by replacing a word and injecting the cross-attention maps of the source image ranging from 0% (left) to 100% (right) of the steps. Without injection, none of the source content is preserved, while injecting throughout all the steps may over-constrain the geometry. The latter results in low fidelity to the text, e.g., the car becomes a bicycle. The full figure is in the appendix (fig. 11).*

attention map layers corresponding to the edited object. In practice, we approximate a mask of the edited part and constrain the modification to be applied only in this local region (lines 11-14 in Algorithm 1). To calculate the mask at step $t$, we compute the average attention map $\overline{M}_{t,w}$ (averaged over steps $T, \ldots, t$) of the original word $w$ and the map $\overline{M}^*_{t,w^*}$ of the new word $w*$. We then apply a threshold to produce binary maps, where $B(x) := x > k$ and $k = 0.3$ throughout all our experiments. To support geometry modifications of the object, the edited region should include the silhouettes of both the original and the newly edited object, therefore, our final mask $\alpha$ is a union of the binary maps. Lastly, we use the mask to constrain the editing region (line 13), where $\odot$ denotes an element-wise multiplication.

**Word Swap.** In this case, the user swaps tokens of the original prompt with others, e.g., $\mathcal{P} =$"a big bicycle" to $\mathcal{P}^* =$"a big car". The main challenge is to preserve the original composition while also addressing the content of the new prompt. To this end, we inject the attention maps of the source image into the generation with the modified prompt. However, the proposed attention injection may over-constrain the geometry, especially when a large structural modification, such as "car" to "bicycle", is involved. We address this by suggesting a softer attention constrain:

$$Edit(M_t, M_t^*, t) := \begin{cases} M_t^* & \text{if } t < \tau \\ M_t & \text{otherwise,} \end{cases}$$

where $\tau$ is a timestamp parameter that determines until which step the injection is applied. Note that the composition is determined in the early steps. Therefore, by limiting the number of injection steps, we can guide the composition while allowing the necessary geometry *freedom* for adapting to the new prompt. An illustration is provided in section 4. Another relaxation is to assign a different number of injection steps for the different tokens in the prompt. If the two words are represented using a different number of tokens, we duplicate/average the maps as necessary using an alignment function as described in the next paragraph.

**Prompt Refinement.** In another setting, the user adds new tokens to the prompt, e.g., $\mathcal{P} =$"a castle" to $\mathcal{P}^* =$"children drawing of a castle". To preserve the common details, we apply the attention injection only over the common tokens from both prompts. Formally, we use an alignment function $A$ that receives a token index from target prompt $\mathcal{P}^*$ and outputs the corresponding token index in $\mathcal{P}$ or *None* if there isn't a match. Then, the editing function is:

$$(Edit\,(M_t, M_t^*, t))_{i,j} := \begin{cases} (M_t^*)_{i,j} & \text{if } A(j) = None \\ (M_t)_{i,A(j)} & \text{otherwise.} \end{cases}$$

Recall that the index $i$ corresponds to a pixel value, where $j$ corresponds to a text token. Again, we may control the number of injection steps. This enables diverse capabilities such as stylization, specification of object attributes, or global manipulations as demonstrated in section 4.

**Attention Re–weighting.** Lastly, the user may wish to strengthen or weakens the extent to which each token affects the resulting image. For example, consider the prompt $\mathcal{P} =$ "a fluffy ball", and assume we want to make the ball more or less fluffy. To achieve such a manipulation, we scale the attention map of the assigned token $j^*$ with a parameter $c \in [-2, 2]$, resulting in a stronger/weaker effect. The rest of the attention maps remain unchanged. The editing function is therefore:

$$(Edit\,(M_t, M_t^*, t))_{i,j} := \begin{cases} c \cdot (M_t)_{i,j} & \text{if } j = j^* \\ (M_t)_{i,j} & \text{otherwise.} \end{cases}$$

As described in section 4, the parameter $c$ allows fine and intuitive control over the induced effect. In addition, since fine textures are generated during the super-resolution phase, we observe that this application can benefit from applying our method also to the super-resolution diffusion model in the case of amplifying or attenuating such fine textures, such as "fluffiness" as shown in fig. 7.

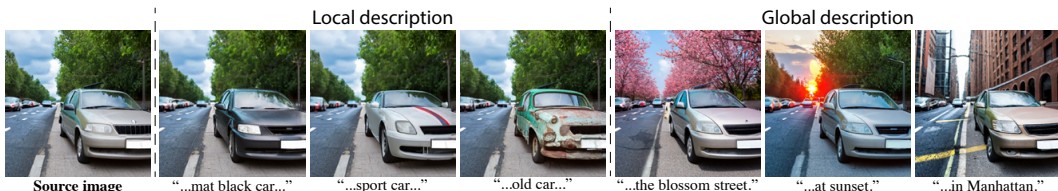

"A car on the side of the street."

**Figure 6: Editing by prompt refinement.** *By extending the description of the initial prompt, we perform local or global editing. Additional results are in the appendix (fig. 13, 23).*

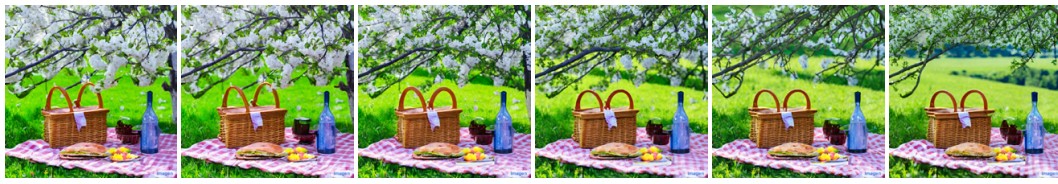

"The picnic is ready under a blossom(↓) tree."

**Figure 7: Text-based editing with fader control.** *By reducing or increasing the cross-attention of specific words (marked with an arrow), we control the extent to which it influences the generation. Additional results are in the appendix (fig. 24).*

### 3.3 SELF-ATTENTION

Most models also consist of self-attention layers, which affect the spatial layout and geometry of the generated image as well. However, unlike cross-attention, the interaction that occurs in *self-attention* layers is only between the pixels to themselves. Therefore, manipulations with respect to specific textual tokens are not feasible. For example, our proposed attention re-weighting and local editing require the matching between the cross-attention maps to the prompt tokens. Another example is presented in the appendix (fig. 16), where we do not inject the attention of the entire prompt but only the attention of a specific word – "butterfly". This enables the preservation of the original butterfly while changing the rest of the content. Contrarily, we can't specify which object should be preserved using only self-attention. Moreover, we observe that the self-attention maps provide inferior semantic control compared to the cross-attention. For instance, as demonstrated in the appendix (fig. 17), using cross-attention injection we can swap between apples and oranges by swapping these words in the prompt. The same experiment fails when using self-attention which lacks a strong interaction between textual tokens and pixels.

Yet, we find that injecting self-attention through a small portion (20%) of the steps in addition to cross-attention injection might further help preserve the source content in some cases. And so, we consider it as an additional tool for Prompt-to-Prompt editing. We provide further analysis in appendix B.

## 4 RESULTS

In this section, we show several applications of our approach and compare it to other methods.

### 4.1 APPLICATIONS

**Text-Only Localized Editing.** We first demonstrate localized editing by modifying the user-provided prompt without requiring any user-provided mask. In fig. 2, we generate an image using the prompt "lemon cake". Our method allows us to retain the spatial layout, geometry, and semantics when replacing the word "lemon" with "apple" (top row). Observe that the background is well-preserved, including the top-left lemons transforming into apples. On the other hand, naively feeding the model with the prompt "apple cake" results in a completely different geometry (2nd row), even when using the same randomness in a deterministic setting (DDIM). Our method succeeds even for a challenging "pasta cake." — the generated cake consists of pasta layers with tomato sauce on top. In case the user adds a new specification, we keep the attention maps of the original prompt, while allowing the generator to address the newly added words. For example, see fig. 6, where we add "old" to the "car", resulting in newly added details over the source car while the background is preserved. Additional results are in the appendix (fig. 16, 22, and 23).

As presented in fig. 5, our method is not confined to modifying only textures and can modify the structure as well, e.g., changing a "bicycle" to a "car". We first show the results without cross-attention injection, where changing a word leads to an entirely different outcome. We then show

Table 1: **User Study results.** *The participants were asked to rate: (1) background / structure preservation with respect to the source image, (2) alignment to the text, and (3) realism.*

|  | VQGAN+CLIP | Text2Live | baseline | Ours |
|---|---|---|---|---|
| (1) Background / Structure ↑ | $1.84 \pm 1.11$ | $4.15 \pm 1.09$ | $3.38 \pm 1.12$ | $4.64 \pm 0.64$ |
| (2) Text Alignment ↑ | $2.46 \pm 1.16$ | $2.89 \pm 1.22$ | $4.26 \pm 1.03$ | $4.55 \pm 0.71$ |
| (3) Realism ↑ | $1.32 \pm 0.70$ | $2.36 \pm 1.12$ | $4.11 \pm 0.93$ | $4.42 \pm 0.82$ |

the resulting image by injecting attention to an increasing number of steps. Note that applying the cross-attention injection in a larger number of steps results in greater similarity to the source image. Therefore, the optimal result is not necessarily achieved by applying the injection throughout all steps. This enables us an even better control by changing the number of injection steps.

**Global editing.** Preserving the composition is not only valuable for local editing, but also an important aspect of global editing. In this setting, the editing should affect all parts of the image, but still retain the original composition, such as the location and identity of the objects. For example, in fig. 6, we preserve the content while changing the lighting. Additional examples are in the appendix (fig. 18), including translating a sketch into a realistic image and inducing an artistic style.

**Fader Control using Attention Re-weighting.** While controlling the image by editing the prompt is very effective, we find that it still does not allow full control over the generated image. Consider the prompt "snowy mountain". A user may want to control the *amount* of snow on the mountain. However, it is quite difficult to describe the desired amount of snow through text. Instead, we suggest a *fader* control (Lample et al., 2017), where the user controls the magnitude of the effect induced by a specific word, as in fig. 7. As described in section 3.2, we achieve such control by re-scaling the attention of the specified word. Additional results are in the appendix (fig. 24, 27 and 30).

**Different Backbone** We use the Imagen (Saharia et al., 2022b) model as a backbone for most of our experiments and results, exploiting its state-of-the-art synthesis quality. However, our method is not limited to a specific model and can be applied to different models as long as they consist of cross-attention layers which are widely used. To validate this, we present results in the appendix (fig. 25, 26, 27, 28, 29, and 30) using the public and popular Latent Diffusion and Stable Diffusion models (Rombach et al., 2021). As can be seen, our method works well using these models as a backbone, enabling various editing capabilities while preserving the source image content. Further analysis is provided in appendix C.1.

**Real Image Editing.** Editing a real image requires finding an initial noise vector that produces the given input image when fed into the diffusion process. This process, known as *inversion*, has recently drawn considerable attention for GANs (Xia et al., 2021b; Bermano et al., 2022), but has not yet been fully addressed for text-guided diffusion models. We show preliminary editing results on real images, based on common inversion techniques for diffusion models. First, a rather naïve approach is to add Gaussian noise to the input image, and then perform a predefined number of diffusion steps. Since this results in significant distortions, we adopt an improved inversion approach (Dhariwal & Nichol, 2021; Song et al., 2020), which is based on the deterministic DDIM model rather than the DDPM. We perform the diffusion process in the reverse direction, that is $x_0 \rightarrow x_T$ instead of $x_T \rightarrow x_0$, where $x_0$ is set to be the given real image. This process often produces satisfying results, as presented in the appendix (fig. 19). However, the inversion is not sufficiently accurate in other cases, as in fig. 20. This is partially due to a distortion-editability tradeoff, where we recognize that reducing the classifier-free guidance (Ho & Salimans, 2021) parameter (i.e., reducing the prompt influence) improves reconstruction but constrains our ability to perform significant manipulations.

To alleviate this limitation, we propose to restore the unedited regions of the original image using a mask, directly extracted from the attention maps. Note that here the mask is generated with no guidance from the user, as described in the local editing paragraph (section 3.2). As presented in fig. 21, this approach works well even using the naïve DDPM inversion scheme (adding noise followed by denoising). Note that the cat's identity is well-preserved under various editing operations, while the mask is produced only from the prompt itself.

## 4.2 COMPARISONS

To evaluate our method, we first randomly generate text-based editing examples from predefined text templates, see appendix F for more details. Source text is then fed to the Imagen model to obtain the source image. We compare our results to other text-guided editing methods: (1) *VQGAN+CLIP*

"Photo of a ~~squirrel~~ **bear** enjoys at the playground."

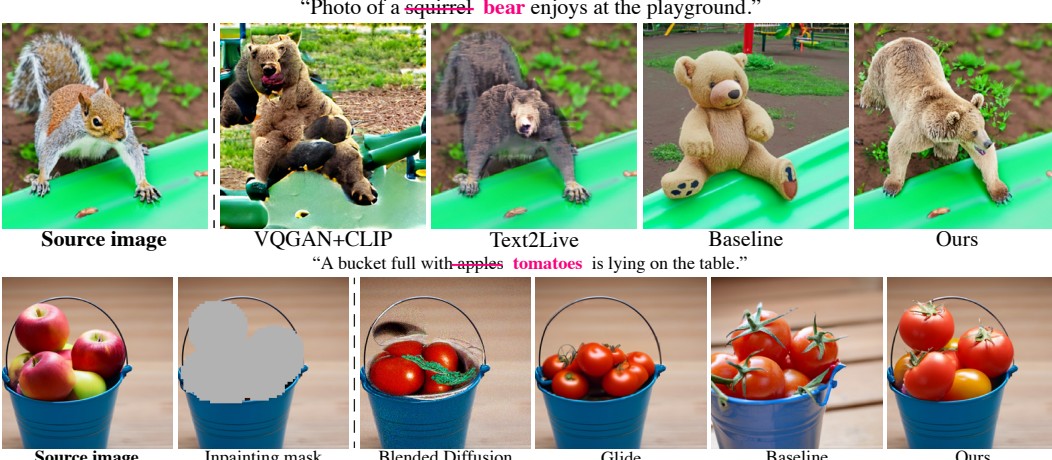

| **Source image** | VQGAN+CLIP | Text2Live | Baseline | Ours |
|---|---|---|---|---|

"A bucket full with ~~apples~~ **tomatoes** is lying on the table."

| **Source image** | Inpainting mask | Blended Diffusion | Glide | Baseline | Ours |
|---|---|---|---|---|---|

Figure 8: **Visual comparison.** *Top: text-guided editing methods (same supervision as ours). Bottom: text-guided inpainting methods which rely on an additional input mask (on the left).*

(Crowson, 2021), (2) *Text2Live* (Bar-Tal et al., 2022), (3) *Blended Diffusion* Avrahami et al. (2022b) and (4) *Glide* (Nichol et al., 2021). We also consider (5) a *baseline* approach where we only replace the source prompt with the target prompt after 20% of diffusion steps using the same random seed.

**Qualitative Comparison.** As can be seen in fig. 8, both VQGAN+CLIP and Text2Live may result in severe artifacts when editing highly structured objects, e.g., a squirrel to a bear. Our method and Text2Live better preserve the background since both methods estimate a mask editing layer. In contrast, the baseline approach produces realistic and meaningful results, but fails to preserve the background. Furthermore, both VQGAN+CLIP and Text2Live require optimization per example which takes 3 and 9 minutes respectively on a GPU. Our method is applied in a single diffusion pass which takes up to 20 seconds.

We also consider text-driven inpainting methods which rely on a given user-defined mask. As can be seen in fig. 8, Glide and Blended Diffusion do produce meaningful edits, but fail to preserve the original structure. Note that these approaches are limited to local changes and cannot handle global edits such as changing the weather in the image. See fig. 14 and 15 in the appendix for more qualitative comparisons.

**Quantitative Comparison.** In the absence of ground truth for text-based editing, quantitative evaluation remains an open challenge. Therefore, similar to (Bar-Tal et al., 2022), we present a user study in table 1. The participants were asked to rate each result in terms of (1) background and structure preservation with respect to the source image, (2) alignment to the text, and (3) realism. Please see appendix E for more details. As shown, the users preferred our method with regard to all three aspects. Glide and Blended Diffusion were not quantitatively evaluated since they require manual labor to produce the input masks.

We provide additional measures in the appendix (table 2) to further validate our claims. We evaluate text-image correspondence using their CLIP score, demonstrating competitive results to methods that directly optimize this metric. In addition, we evaluate the perceptual similarity between the original and edited images using LPIPS (Zhang et al., 2018a) and MS-SSIM (Wang et al., 2003). This shows our capability of performing *local* editing, similar to Text2Live (Bar-Tal et al., 2022). However, CLIP score and perceptual similarity do not reflect our superior quality and realism which are demonstrated in the user study.

## 5 CONCLUSIONS

In this work, we uncovered the powerful capabilities of the cross-attention layers within text-to-image diffusion models. We showed that these high-dimensional layers have an interpretable representation of spatial maps that play a key role in tying the words in the text prompt to the spatial layout of the synthesized image. With this observation, we showed how various manipulations of the prompt can directly control attributes in the synthesized image, paving the way to various applications including local and global editing. This work is a first step towards providing users with simple and intuitive means to *edit* images and navigate through a semantic, textual, space, which exhibits incremental changes after each step, rather than producing an image from scratch after each text manipulation.

## 6 ETHIC STATEMENT

Our work suggests a new editing technique for images that are generated using state-of-the-art text-to-image diffusion models. As explained in section 4.1 and appendix D, our approach can edit real images, although this still remains a more challenging setting. Such manipulation of real photos might be exploited by malicious parties to produce fake content in order to spread disinformation. This is a known problem, common to all image editing techniques. However, research in identifying and preventing malicious editing is already making significant progress. We believe our work would contribute to this line of work, since we provide a comprehensive analysis of the editing procedure using text-to-image diffusion models.

## ACKNOWLEDGMENTS

We thank Noa Glaser, Adi Zicher, Yaron Brodsky, Shlomi Fruchter and David Salesin for their valuable inputs that helped improve this work, and to Mohammad Norouzi, Chitwan Saharia and William Chan for providing us with their support and the pretrained models of Imagen (Saharia et al., 2022b). Special thanks to Yossi Matias for early inspiring discussion on the problem and for motivating and encouraging us to develop technologies along the avenue of intuitive interaction.

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

# A  BACKGROUND

## A.1  DIFFUSION MODELS

Diffusion Denoising Probabilistic Models (DDPM) Sohl-Dickstein et al. (2015); Ho et al. (2020) are generative latent variable models that aim to model a distribution $p_\theta(x_0)$ that approximates the data distribution $q(x_0)$ and easy to sample from. DDPMs model a "forward process" in the space of $x_0$ from data to noise.[1] This process is a Markov chain starting from $x_0$, where we gradually add noise to the data to generate the latent variables $x_1, \ldots, x_T \in X$. The sequence of latent variables therefore follows $q(x_1, \ldots, x_t \mid x_0) = \prod_{i=1}^{t} q(x_t \mid x_{t-1})$, where a step in the forward process is defined as a Gaussian transition $q(x_t \mid x_{t-1}) := N(x_t; \sqrt{1-\beta_t} x_{t-1}, \beta_t I)$ parameterized by a schedule $\beta_0, \ldots, \beta_T \in (0, 1)$. When $T$ is large enough, the last noise vector $x_T$ nearly follows an isotropic Gaussian distribution.

An interesting property of the forward process is that one can express the latent variable $x_t$ directly as the following linear combination of noise and $x_0$ without sampling intermediate latent vectors:

$$x_t = \sqrt{\alpha_t} x_0 + \sqrt{1 - \alpha_t} w, \ \ w \sim N(0, I), \tag{2}$$

where $\alpha_t := \prod_{i=1}^{t} (1 - \beta_i)$.

In order to sample from the distribution $q(x_0)$, we define the dual "reverse process" $p(x_{t-1} \mid x_t)$ from isotropic Gaussian noise $x_T$ to data by sampling the posteriors $q(x_{t-1} \mid x_t)$. Since the intractable reverse process $q(x_{t-1} \mid x_t)$ depends on the unknown data distribution $q(x_0)$, we approximate it with a parameterized Gaussian transition network $p_\theta(x_{t-1} \mid x_t) := N(x_{t-1} \mid \mu_\theta(x_t, t), \Sigma_\theta(x_t, t))$. The $\mu_\theta(x_t, t)$ can be replaced (Ho et al., 2020) by predicting the noise $\epsilon_\theta(x_t, t)$ added to $x_0$ using equation 2.

Under this definition, we use Bayes' theorem to approximate

$$\mu_\theta(x_t, t) = \frac{1}{\sqrt{\alpha_t}} \left( x_t - \frac{\beta_t}{\sqrt{1 - \alpha_t}} \epsilon_\theta(x_t, t) \right). \tag{3}$$

Once we have a trained $\epsilon_\theta(x_t, t)$, we can using the following sample method

$$x_{t-1} = \mu_\theta(x_t, t) + \sigma_t z, \ \ z \sim N(0, I). \tag{4}$$

We can control $\sigma_t$ of each sample stage, and in DDIMs (Song et al., 2020) the sampling process can be made deterministic using $\sigma_t = 0$ in all the steps. The reverse process can finally be trained by solving the following optimization problem:

$$\min_\theta L(\theta) := \min_\theta E_{x_0 \sim q(x_0), w \sim N(0, I), t} \left\| w - \epsilon_\theta(x_t, t) \right\|_2^2,$$

teaching the parameters $\theta$ to fit $q(x_0)$ by maximizing a variational lower bound.

## A.2  ATTENTION LAYERS IN TEXT TO IMAGE DIFFUSION MODELS

We implement our method on three different diffusion models: Imagen, Latent Diffusion, and Stable Diffusion. We describe here only a high-level description of each model and its attention layers that are relevant to our method. Note that these models condition on the text prompt in the noise prediction of each diffusion step through two types of attention layers: i) cross-attention layers. ii)

---

[1] This process is called "forward" due to its procedure progressing from $x_0$ to $x_T$.

hybrid attention that acts both as self-attention and cross-attention by concatenating the text embedding sequence to the key-value pairs of each self-attention layer. Our method only intervenes in the cross-attention part of the hybrid attention. That is, only the last channels, which refer to text tokens, are modified in the hybrid attention modules.

**Imagen.** (Saharia et al., 2022b) consists of three text-conditioned diffusion models and a language model: A text-to-image $64 \times 64$ model, two super-resolution models – $64 \times 64 \rightarrow 256 \times 256$ and $256 \times 256 \rightarrow 1024 \times 1024$ and a pre-trained T5 XL language model Raffel et al. (2020). These predict the noise $\epsilon_\theta(z_t, c, t)$ via a U-shaped network, for $t$ ranging from $T$ to 1. Where $z_t$ is the latent vector and $c$ is the text embedding of the language model. We highlight the differences between the three diffusion models:

- $64 \times 64$ – starts from a random noise, and uses the U-Net as in (Dhariwal & Nichol, 2021). This model is conditioned on text embeddings via both cross-attention layers at resolutions $[16, 8]$ and hybrid-attention layers at resolutions $[32, 16, 8]$ of the downsampling and upsampling within the U-Net.

- $64 \times 64 \rightarrow 256 \times 256$ – conditions on a naively upsampled $64 \times 64$ image. An efficient version of a U-Net is used, which includes Hybrid attention layers in the bottleneck (resolution of 32).

- $256 \times 256 \rightarrow 1024 \times 1024$ – conditions on a naively upsampled $256 \times 256$ image. An efficient version of a U-Net is used, which only includes cross-attention layers in the bottleneck (resolution of 64).

**Latent Diffusion.** Latent Diffusion Model (LDM) (Rombach et al., 2021) is substantially different from Imagen. First, to reduce memory consummation, LDM operates in the latent space of a pre-trained VQGAN Yu et al. (2021); Esser et al. (2021a). This reduces the spatial size of an input image from $256 \times 256$ to a quantized latent space of size $32 \times 32$ with 4 channels. Second, the language model is trained from scratch with the main diffusion model and consists of 32 transformer layers. For the diffusion process, a U-Net is used as in (Dhariwal & Nichol, 2021), which consists of self-attention layers followed by text-conditioned cross-attention layers at resolutions $32, 16, 8$ and $4$.

**Stable Diffusion.** Stable Diffusion (SD) is an improved version of LDM which is trained on higher resolution with more resources and data. The latent space is of size $64 \times 64$ with 4 channels, which after decoding results in an image of size $512 \times 512$. SD uses pre-trained CLIP model (Radford et al., 2021) for the conditioned text embedding, and consists of self-attention layers followed by text-conditioned cross-attention layers at resolutions $64, 32, 16$ and $8$.

## B    SELF-ATTENTION IN TEXT-CONDITIONED DIFFUSION MODELS

An interesting question is the role of self-attention maps. In particular, compared to cross-attention, how well it reveals the structure of the generated image, and how its injection affects the image generation under our settings.

Similar to the cross-attention maps, we found that self-attention maps are correlated to the structure and different semantic regions in the image. As can be seen in fig. 9, the self-attention maps of different pixels highlight the close region of the pixel in addition to regions in the image that contain the same semantic content. For example, a pixel on the crust of the pizza attends to other pixels on the crust. In addition, if we look at the top principle components of the self-attention maps, we can clearly identify the layout of the generated image, as previously shown in (Tumanyan et al., 2022) for a different model. However, since the self-attention maps are not correlated to specific words, these provide inferior control compared to cross-attention maps. For instance, it is much more challenging to find a map that highlights only the pepperoni using self-attention.

Next, we inject the self-attention maps of a source image during the generation of an image conditioned on another target prompt. Notice that the source and target prompts might be unaligned in this scenario. Such examples are shown in fig. 12, where we apply self-attention injection for a gradually increased number of diffusion steps. As we can see, the attention maps drastically affect the resulting images such that injecting the maps for more than $50\%$ steps suppresses almost any connection to the target prompt. Interestingly, the self-attention maps can also determine the color palette in the image. Since the self-attention injection may restrict the editing capability of our

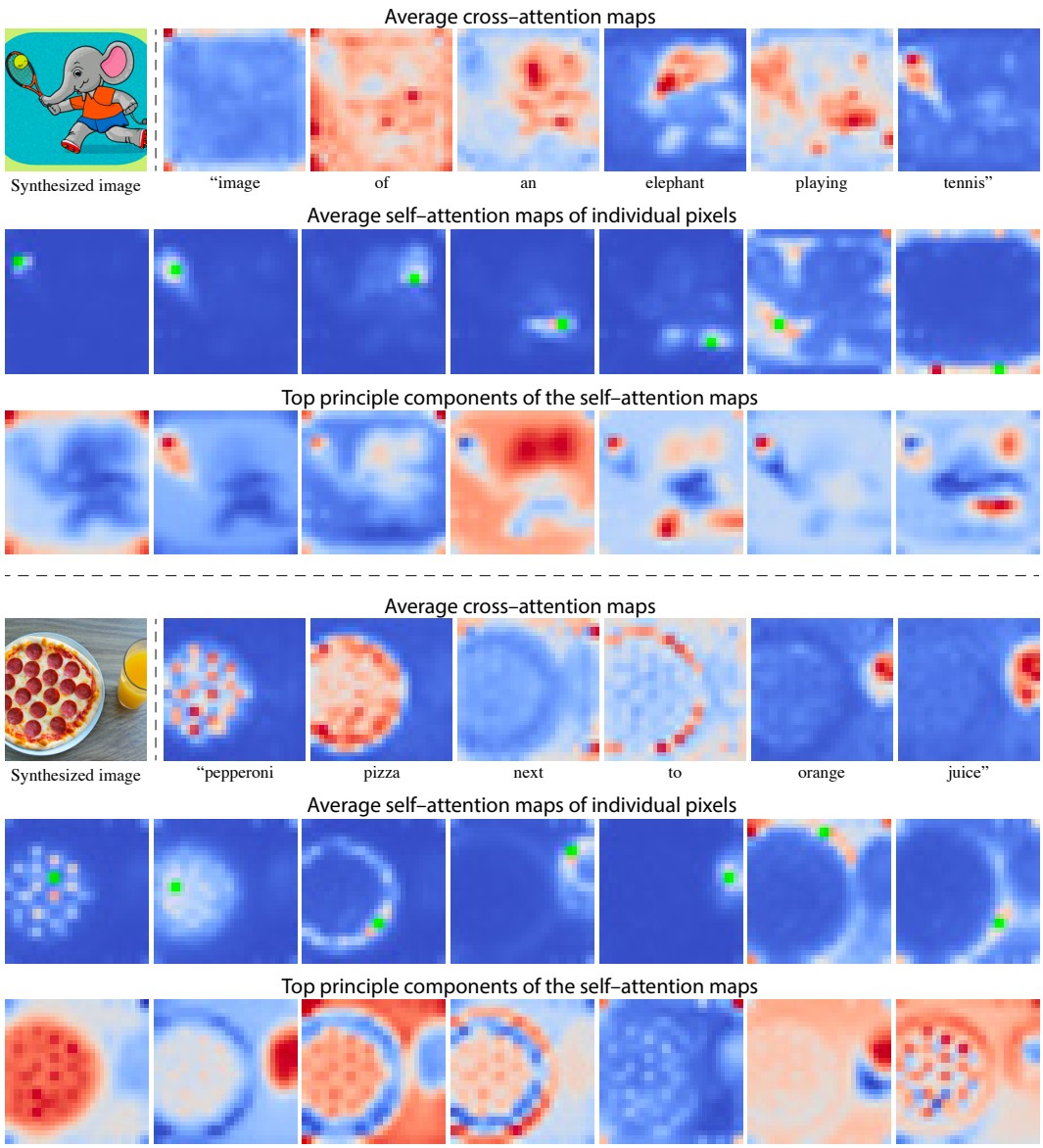

Figure 9: **Visualization of cross and self-attention.** *The top row for each example illustrates the average cross-attention maps for the given prompt. The second row shows the self-attention maps with respect to different pixels (marked in green). The third row shows the principle components of the self-attention maps. All examples present the attention maps at resolution $16 \times 16$ after averaging across diffusion steps, different layers, and attention heads.*

method, we use self-attention injection for up to $20\%$ of the diffusion steps. We found that this may improve the source background preservation in some cases.

## C ADDITIONAL RESULTS

Additional quantitative results are provided in table 2.

Full figures for fig. 5 and 6, are in fig. 11 and 13 respectively. Additional qualitative comparisons are provided in fig. 14 and 15.

In fig. 16, we do not inject the attention of the entire prompt but only the attention of a specific word – "butterfly". This enables the preservation of the original butterfly while changing the rest of the content. As demonstrated in fig. 17, using cross-attention injection we can swap between apples and oranges by swapping these words in the prompt. The same experiment fails when using self-attention which lacks a strong interaction between textual tokens and pixels.

Additional global editing results are presented in fig. 18, illustrating a translation of a sketch into a photo-realistic image and inducing an artistic style. Examples for editing of real images provided in fig. 19, 20, and 21.

We provide additional visual examples for different editing operations using our method: fig. 22 show word swap results, fig. 23 show adding specification to an image, and fig. 24 show attention re-weighting.

### C.1 DIFFERENT BACKBONES

Results for the Latent Diffusion and Stable Diffusion models are in fig. 25, 26, 27, 28, 29, and 30. We observe that text-based replacement and refinement operations work well for all three models. However, we notice a small difference between the three models in the *Fader Control using Attention Re-weighting*. Visual examples of this application are presented in fig. 24, 27 and 30 using Imagen, Latent Diffusion and Stabe Diffusion respectively. As can be seen, when using Imagen as the backbone, our method produces high-quality results and can even handle delicate changes such as reducing the "cubic" appearance of sushi. On the other hand, applying our method with Stable Diffusion may result in unexpected artifacts. For example, when reducing the attention to the word "night" (last example in fig. 30), not only the time of the day is changed but also the dark skies turns into trees.

We hypothesize that this difference is the result of using different language models for text embedding. Imagen uses a T5 language model that is trained using an unsupervised language objective of span masking. Stable Diffusion uses CLIP which is trained with a multi-modal constructive objective. Lastly, the Latent Diffusion language model is trained with the same reconstitution objective as the diffusion model. Therefore, we suggest that the text embedding of T5 better represents disentangled information and so yields superior results.

### D LIMITATIONS

While we have demonstrated semantic control by changing only textual prompts, our technique is subject to a few limitations. First, the current inversion process results in a visible distortion over some of the test images, see fig. 20. Moreover, the inversion requires the user to come up with a suitable prompt which could be challenging for complicated compositions. Note that the challenge of inversion for text-guided diffusion models is an orthogonal endeavor to our work, which would be studied in the future. Second, current attention maps are of low resolution, as the cross-attention is placed in the network's bottleneck. This bounds our ability to perform more precise editing. To alleviate this, we suggest incorporating cross-attention also in higher-resolution layers. We leave this for future work as it requires analyzing the training which is out of our scope. Third, our method requires setting the timestamp parameter for attention injection and the scale parameter for attention re-weighting. Tuning these usually requires roughly a minute for the Imagen model. However, our method is not highly sensitive to these, and using a constant timestamp produces satisfying results in most cases.Furthermore, we believe that future works will reduce the inference time of these models, so tuning the parameter will be quicker and more intuitive. Finally, we recognize that our method cannot be used for large structural changes in the image, like changing the pose of an animal, move objects or changing the number of objects in the image, see examples in fig. 10. We leave this kind of control for future work.

### E USER STUDY

32 participants answered our user study. Each was asked to evaluate 18 randomly selected Prompt-to-Prompt examples for each method. The examples were given in random order and were divided into three parts: (A) consists of 6 replacement examples using templates 1 and 2 (see appendix F). (B) consists of 6 local refinement examples using templates 3 and 4. (C) consist of 6 global refine-

Figure 10: **Editing Failure Cases.** *Since Prompt-to-Prompt preserves the overall structure of the source image, it fails when large structural modification is required. For example, changing the pose of the dog from "playing" to "sleeping" (on the left) or changing the number of chairs (middle). In addition, our method is limited by the semantic understanding of the diffusion model, for example, on the right, we add the refinement "smiling bunny doll" that made both the bunny and the child smile.*

ment examples using templates 5 and 6. For each example the user was asked to rate the image on a $1 - 5$ scale (higher is better) with respect to the following questions:

(1) How well does the right image preserve the structure and the background of the left image? Consider the preservation of properties that are not specified by the text above the images.

(2) How well does the right image match the text description above it? Specifically, consider the highlighted text.

(3) Rate the overall realism and quality of the right image.

See fig. 31 for screenshots.

## F    EVALUATION PROMPTS

We use the following prompt templates to generate the evaluation data:

```
Template_1:  "Image of <A_RPC> inside a <B_CONST>."
Select_A = ["apples", "oranges", "chocolates", "kittens", "puppies", "candies"]
Select_B = ["box", "bowl", "bucket", "nest", "pot"]

Template_2:  "A <A_RPC> full of <B_CONST> is lying on the table."
Select_A = ["box", "bowl", "bucket", "nest", "pot"]
Select_B = ["apples", "oranges", "chocolates", "kittens", "puppies", "candies"]

Template_3:  "Photo of a <A_RPC> <CONST>."
select_from_a = ["cat", "dog", "lion", "camel", "horse", "bear", "squirrel",
                "elephant", "zebra", "giraffe", "cow"]
select_from_b = ["seating in the field", "walking in the field", "walking in the city",
                "wandering around the city", "wandering in the streets", "walking in the desert",
                "seating in the desert", "walking in the forest", "seating in the forest",
                "walking in the desert", "seating in the desert", "plays at the playground",
                "enjoys at the playground"]

Template_3:  "Photo of a <A_CONST> <B_ADD> with a <C_CONST> on it."
Select_A = ["tree", "shrub", "flower", "chair", "fruit"]
select_B = ["made of candies", "made of bricks", "made of paper", "made of clay",
            "made of wax", "made of feathers"]
Select_C = ["bug", "butterfly", "bee", "grasshopper", "bird"]

Template_4:  "Image of a <A_ADD> <B_CONST> on the side of the road."
Select_A = ["wooden", "old", "crashed", "golden", "silver", "sport", "toy"]
select_B = ["car", "bus", "bicycle", "motorcycle", "scooter", "van"]

Template_5:  "A landscape Image of <A_CONST> <B_ADD>."
Select_A = ["a river", "a lake", "a valley", "mountains",
            "a forest", "a river in the valley",
            "a vlilage on a mountain",
            "a waterfall between the mountains", "the cliffs in the desert"]
select_B = ["in the winter", "in the autumn", "at night", "at sunset", "at sunrise",
            "at fall", "in rainy day","in a cloudy day", "at evening"]
```

**Source image**

**Source Prompt:** "Photo of a cat riding on a bicycle."

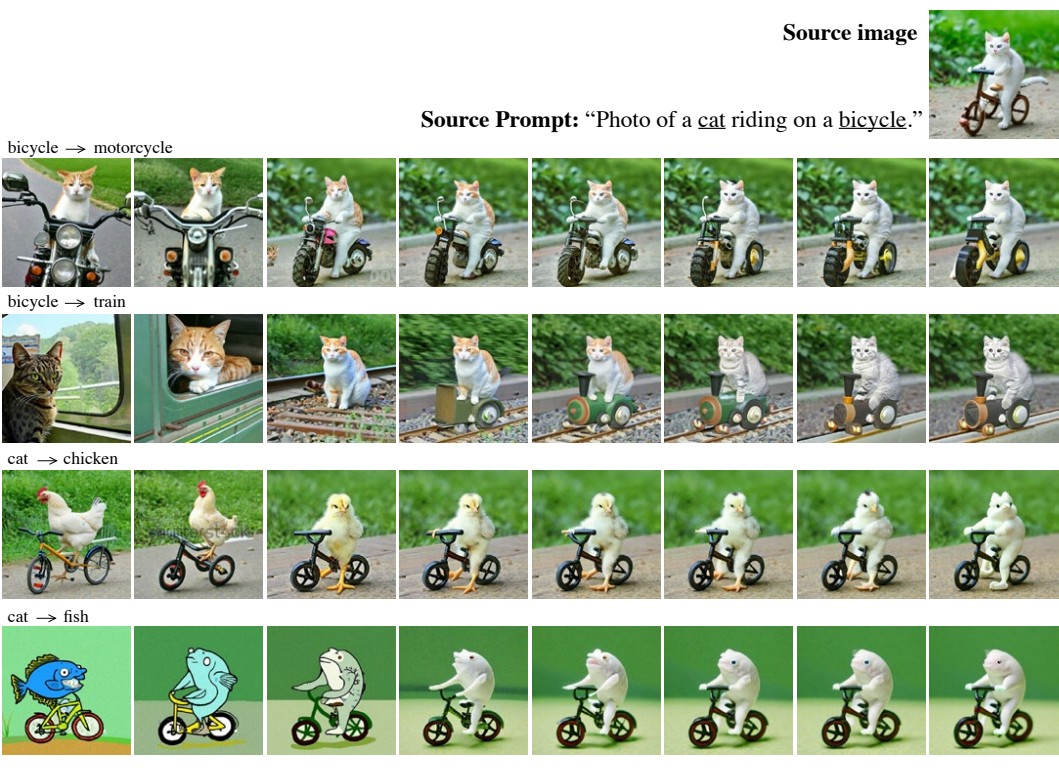

bicycle → motorcycle

bicycle → train

cat → chicken

cat → fish

W.O. cross–attention injection ⟶ Full cross–attention injection

Figure 11: **Attention injection through a varied number of diffusion steps.** *Top: source image and prompt. In each row, we modify the content of the image by replacing a single word in the text and injecting the cross-attention maps of the source image ranging from 0% (left) to 100% (right) of the steps. Without our method, none of the source image content is guaranteed to be preserved. On the other hand, injecting the cross-attention throughout all the steps may over-constrain the geometry, resulting in low fidelity to the text.*

```
Template_6 : "<A_CONST> in the <B_ADD> street."
Select_A = ["Heavy traffic", "The houses", "The buildings", "Cycling",
            "The tram is passing", "The bus arrived at the station"]
select_B = ["snowy", "flooded", "blossom", "modern",
            "historic", "commercial", "colorful"]
```

We generate 20 random examples using each template where the tokens <CONST>, <RPC> and <ADD> where randomly replaced with one item in the corresponding selection list below each template.

<CONST> stands for phrase that is used in both source and target prompt.

<RPC> stands for phrase that is different between the source and target.

<ADD> stands for refinement phrase which is only replaced in the target prompt and omitted in the source prompt.

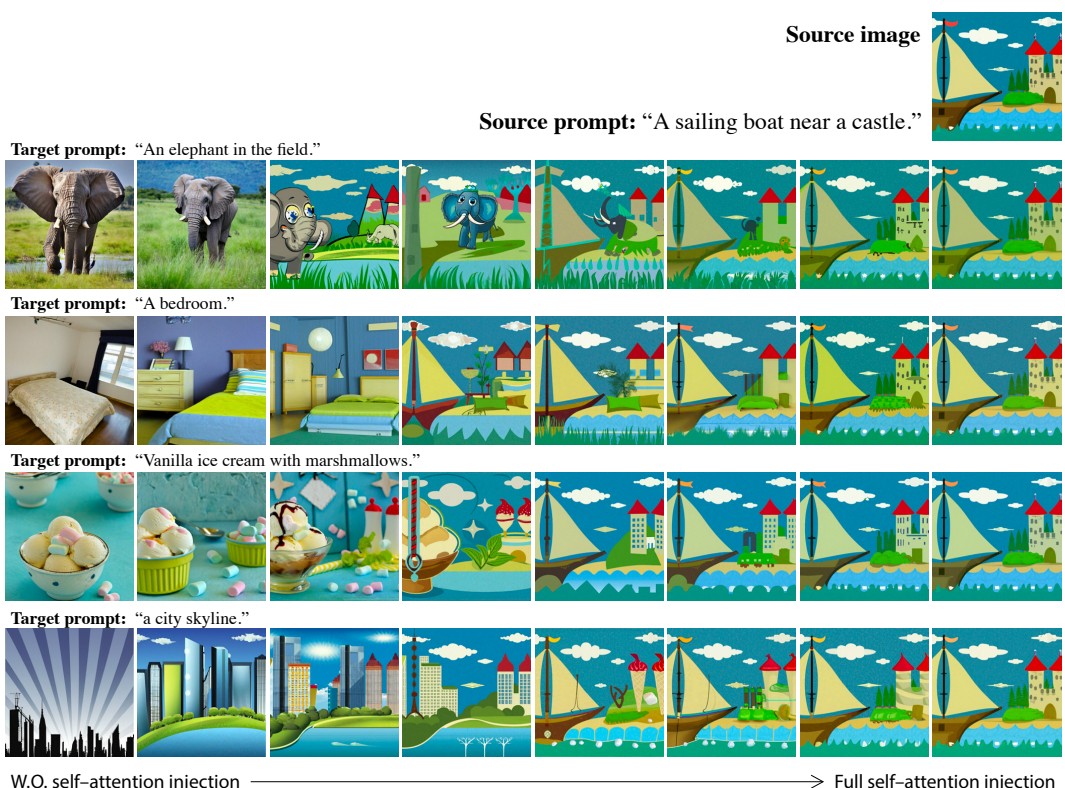

Figure 12: **Self-attention injection through a varied number of diffusion steps.** *In each row, we conditioned the image generation on a new target prompt and inject the self-attention maps of the source image ranging from 0% (left) to 100% (right) of the diffusion steps.*

Table 2: Additional quantitative results. We measure text-image correspondence using CLIP (Radford et al., 2021), demonstrating competitive results to methods that directly optimize the CLIP score. In addition, we evaluate the similarity between the original and the edited images using the LPIPS (Zhang et al., 2018a) perceptual distance and MS-SSIM (Wang et al., 2003). This show our capability of performing *local* editing, similar to Text2Live (Bar-Tal et al., 2022).

|  | CLIP score ↑ | MS-SSIM ↑ | LPIPS ↓ |
|---|---|---|---|
| VQGAN+CLIP | $0.282 \pm 0.04$ | $0.27 \pm 0.046$ | $0.64 \pm 0.05$ |
| Text2Live | $0.247 \pm 0.04$ | $0.82 \pm 0.065$ | $0.25 \pm 0.05$ |
| baseline | $0.253 \pm 0.03$ | $0.69 \pm 0.13$ | $0.35 \pm 0.12$ |
| Ours | $0.253 \pm 0.04$ | $0.81 \pm 0.11$ | $0.22 \pm 0.1$ |

"A car on the side of the street."

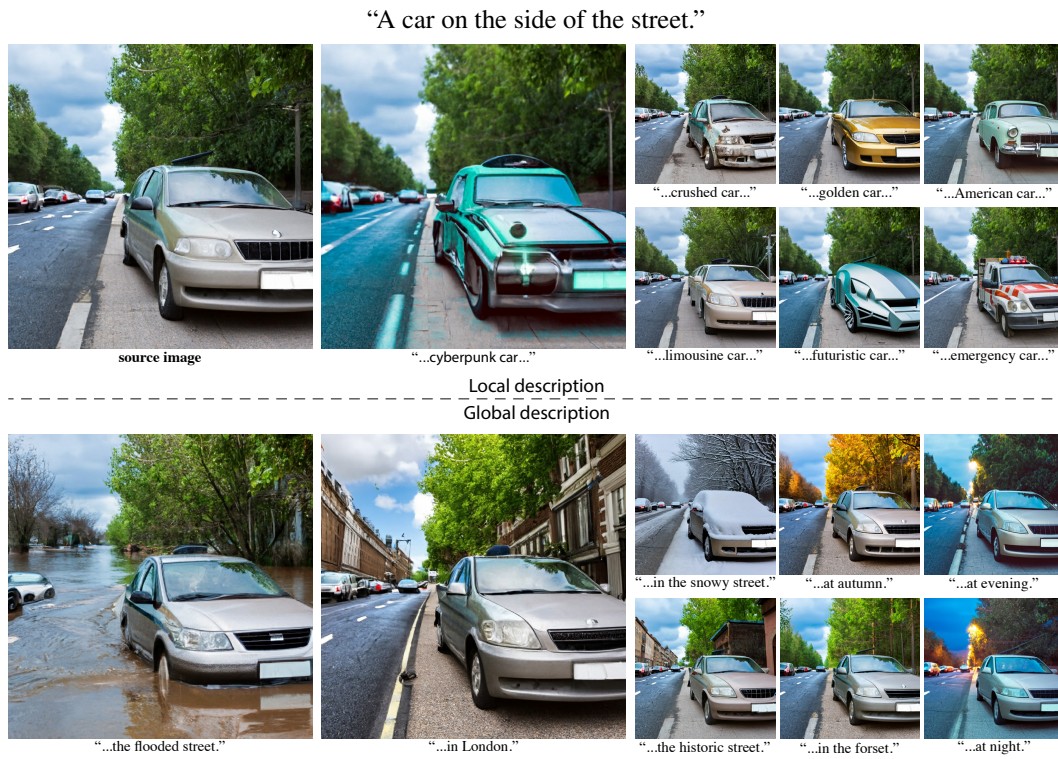

Figure 13: **Editing by prompt refinement.** *By extending the description of the initial prompt, we can make local edits to the car (top rows) or global modifications (bottom rows).*

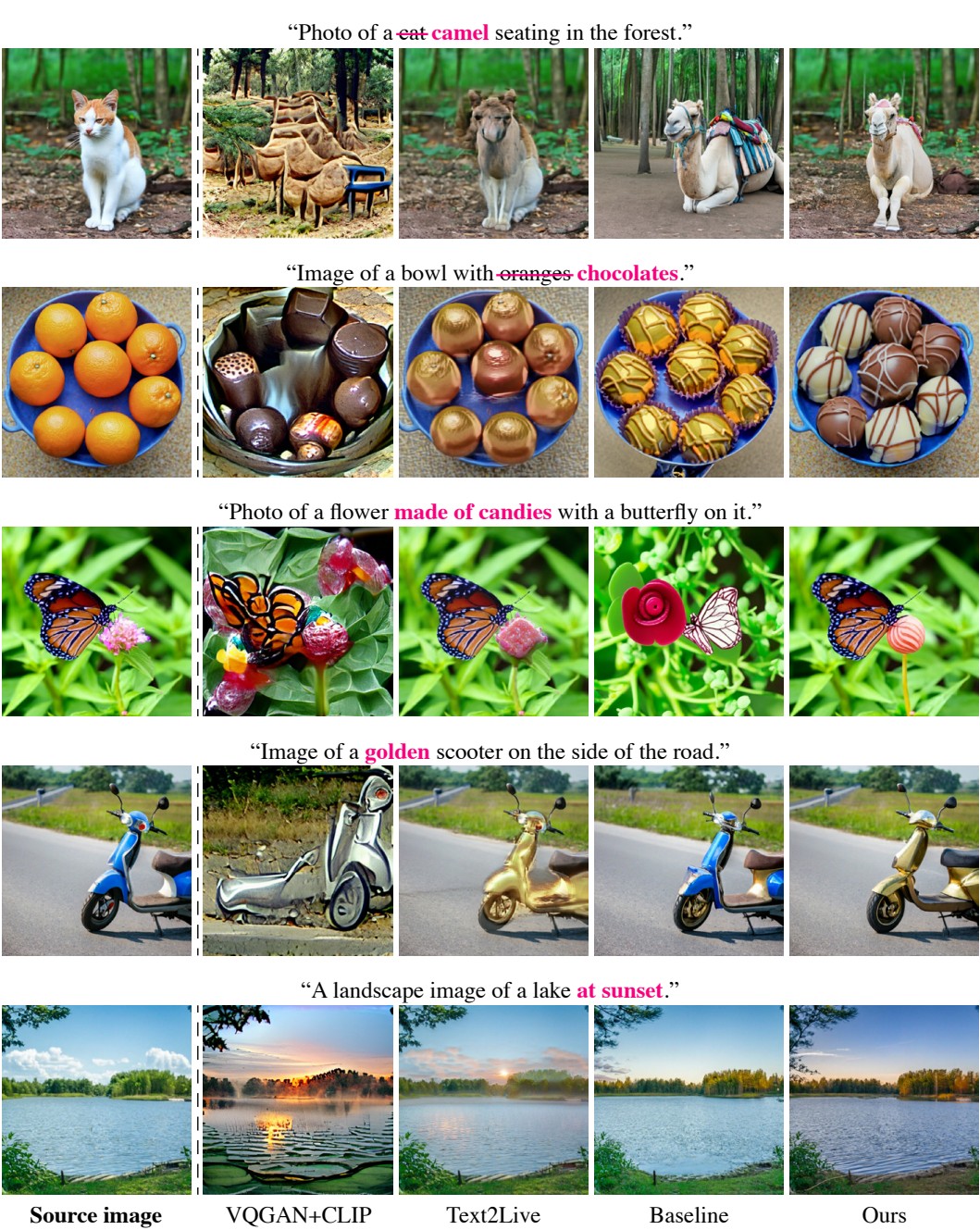

Figure 14: Additional comparisons to text-guided image editing. Similar to ours, these methods do not require a user-provided mask.

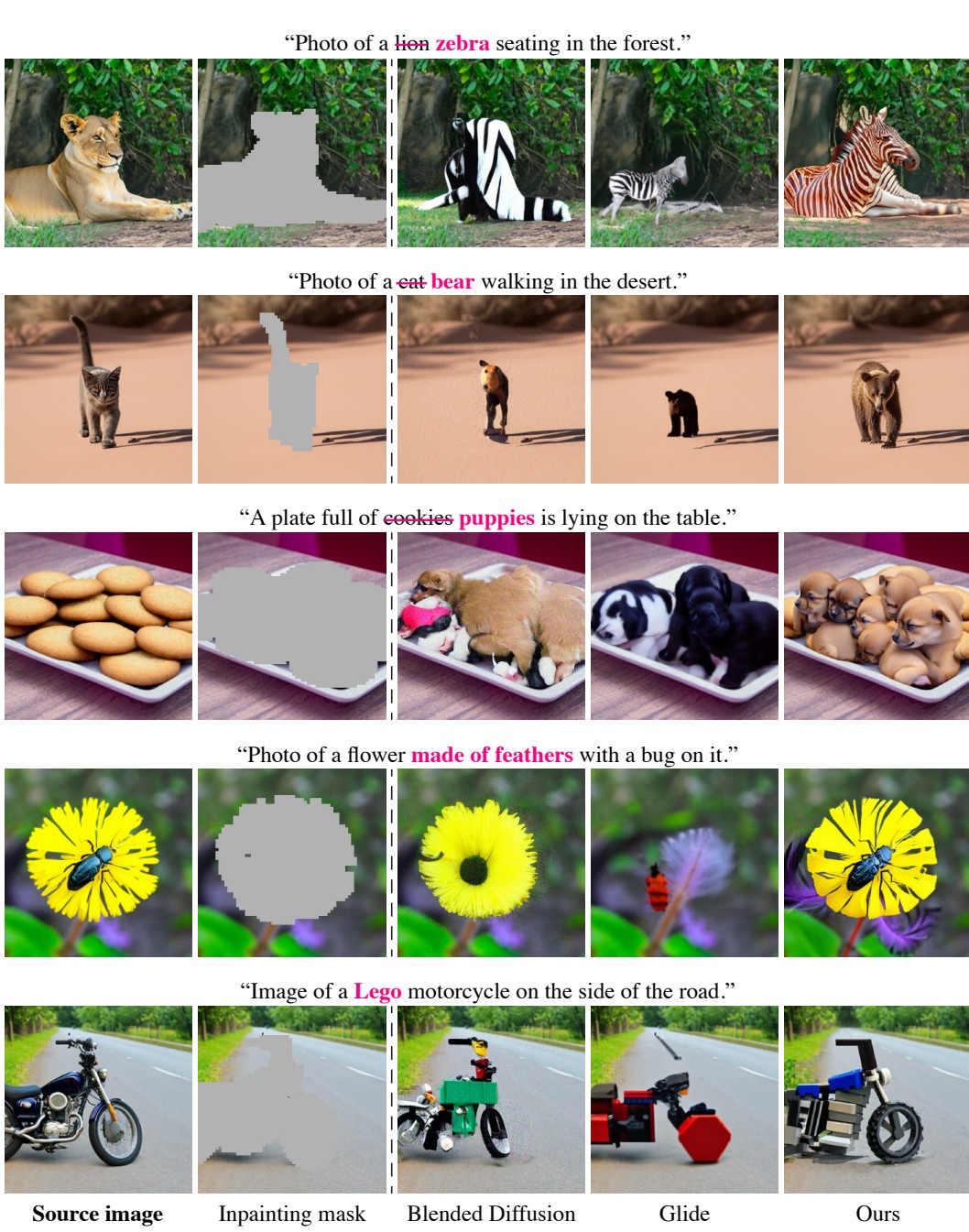

Figure 15: Additional comparisons to text-guided in-painting methods. Unlike our method, these techniques require an auxiliary segmentation mask which is provided by the user.

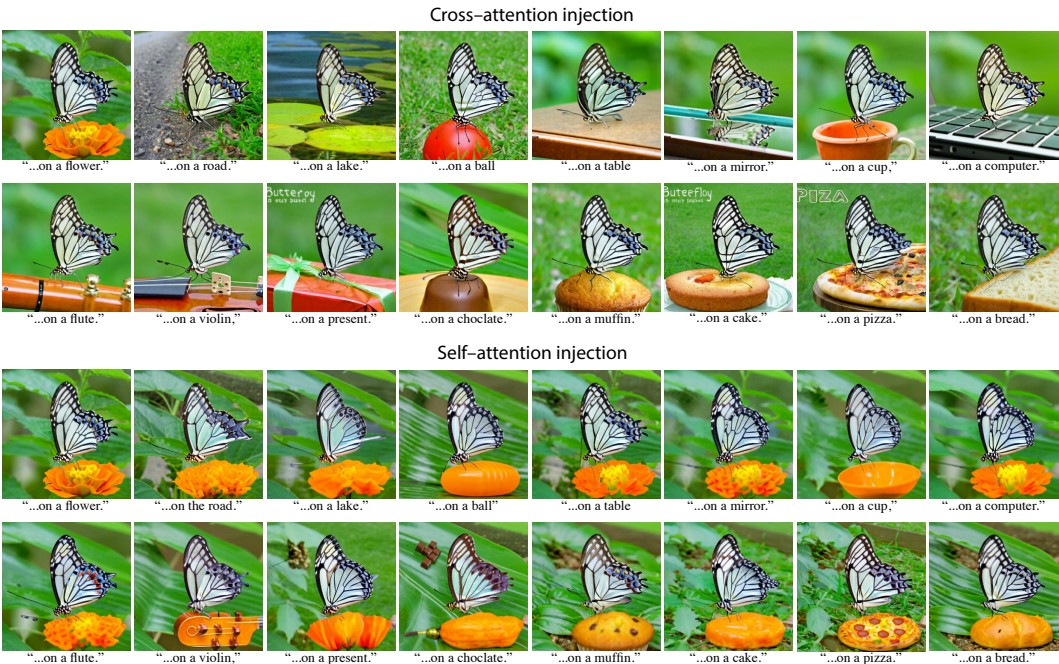

Figure 16: **Object preservation and replacement using cross and self attention injection.** *Top: by injecting only the cross-attention weights of the word "butterfly" taken from the top-left image we can preserve the structure and appearance of a single item while replacing its context (i.e., background). Bottom: using only self-attention injection we can't specify which object should be preserved, therefore, modifying the background while keeping the butterfly is more challenging.*

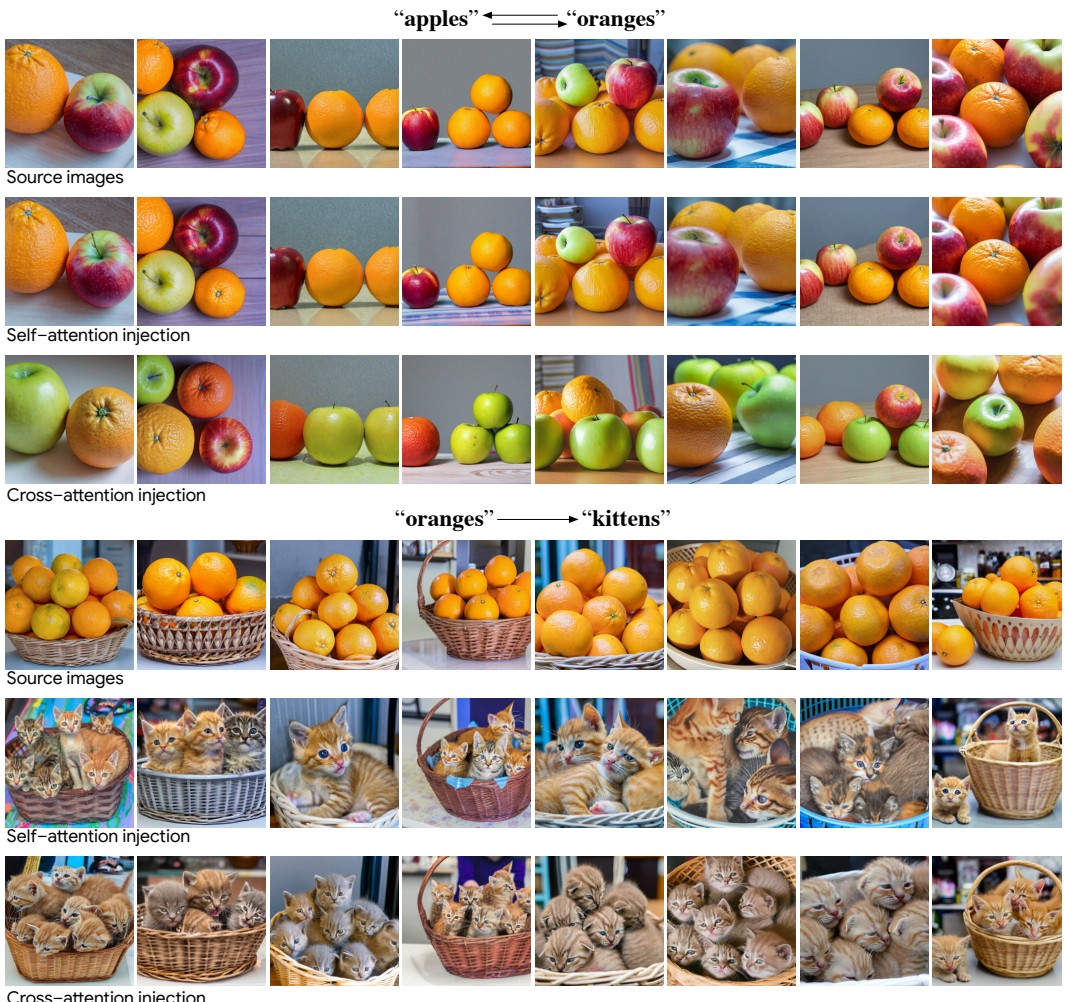

Figure 17: **Object replacement using cross-attention injection and self-attention injection.** *Cross–attention injection better preserves the semantic relation between the generated image and the text prompt. Top: using cross-attention injection (third row) we can swap between apples and oranges in the source image by swapping these words in the prompt "apples and oranges are on the table.". The same experiment fails when using self-attention which lacks a strong interaction between textual tokens and pixels. Bottom: cross-attention injection (6th row) better preserves the distinct elements in the image when replacing the word "oranges" with "kittens" in the sentence "a basket with oranges on the counter."*

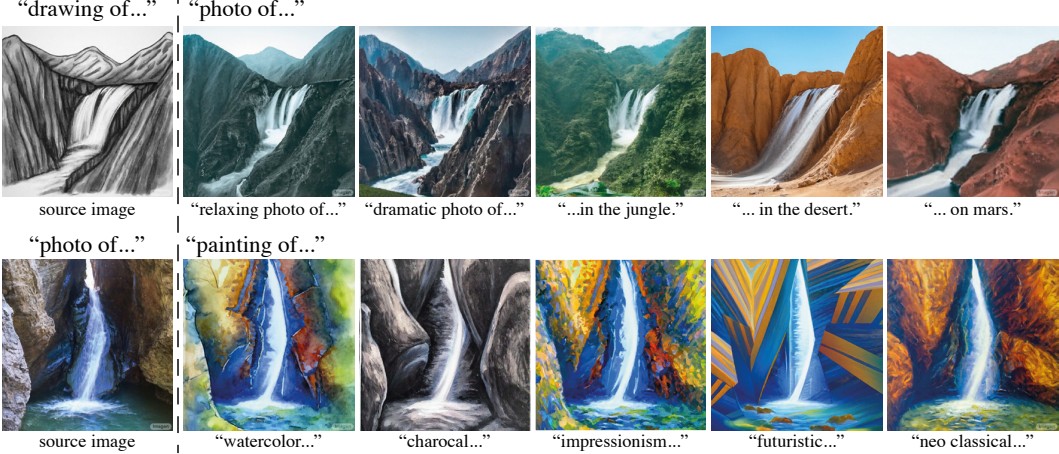

Figure 18: **Image stylization.** *By adding a style description to the prompt while injecting the source attention maps, we can create various images in the new desired styles that preserve the structure of the original image.*

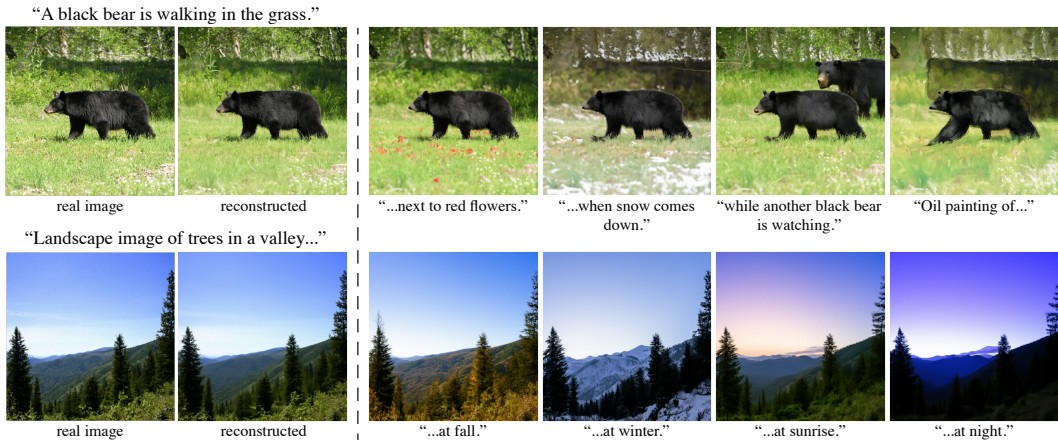

Figure 19: **Editing of real images.** *On the left, inversion results using DDIM Song et al. (2020) sampling. We reverse the diffusion process initialized on a given real image and text prompt. This results in a latent noise that produces an approximation to the input image when fed to the diffusion process. Afterward, on the right, we apply our Prompt-to-Prompt technique to edit the images.*

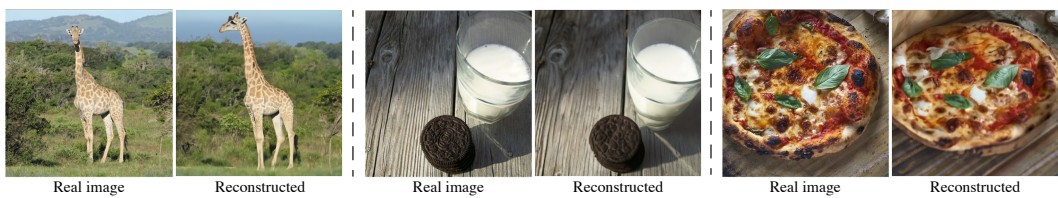

Figure 20: Inversion Failure Cases. Current DDIM-based inversion of real images might result in unsatisfied reconstructions.

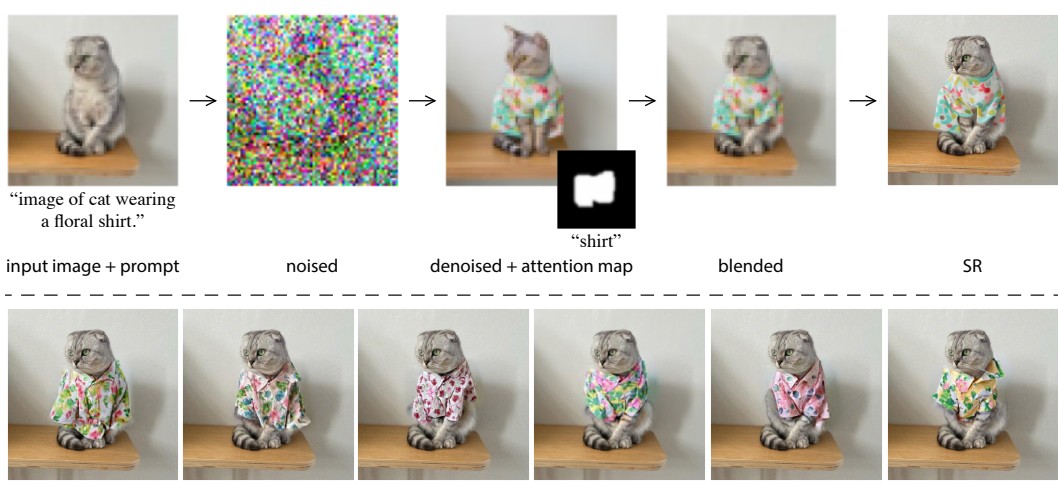

Figure 21: **Mask-based editing.** *Using the attention maps, we preserve the unedited parts of the image when the inversion distortion is significant. This does not require any user-provided masks, as we extract the spatial information from the model using our method. Note how the cat's identity is retained after the editing process.*

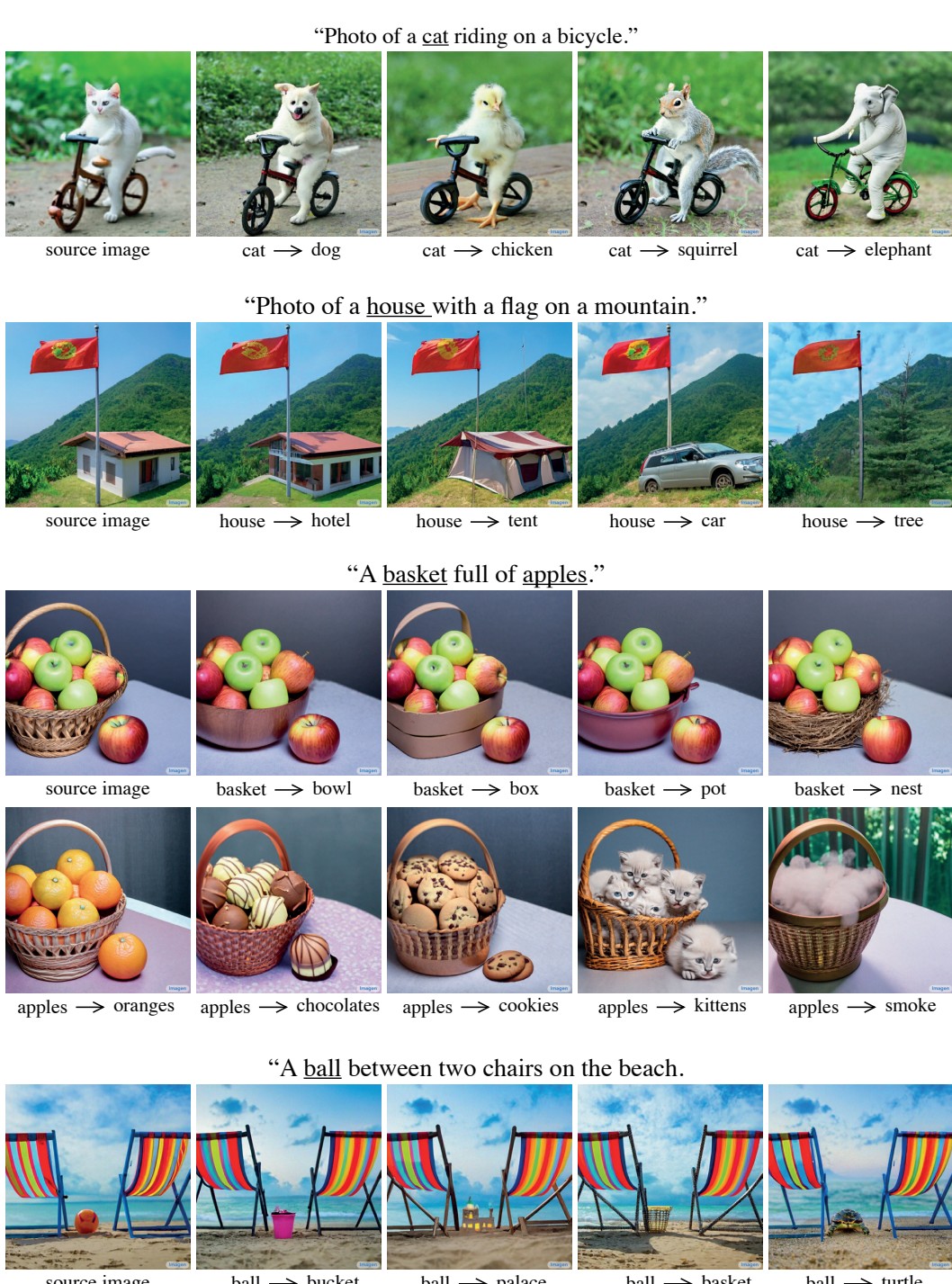

Figure 22: Additional results for Prompt-to-Prompt editing by word swapping using the Imagen model (Saharia et al., 2022b)..

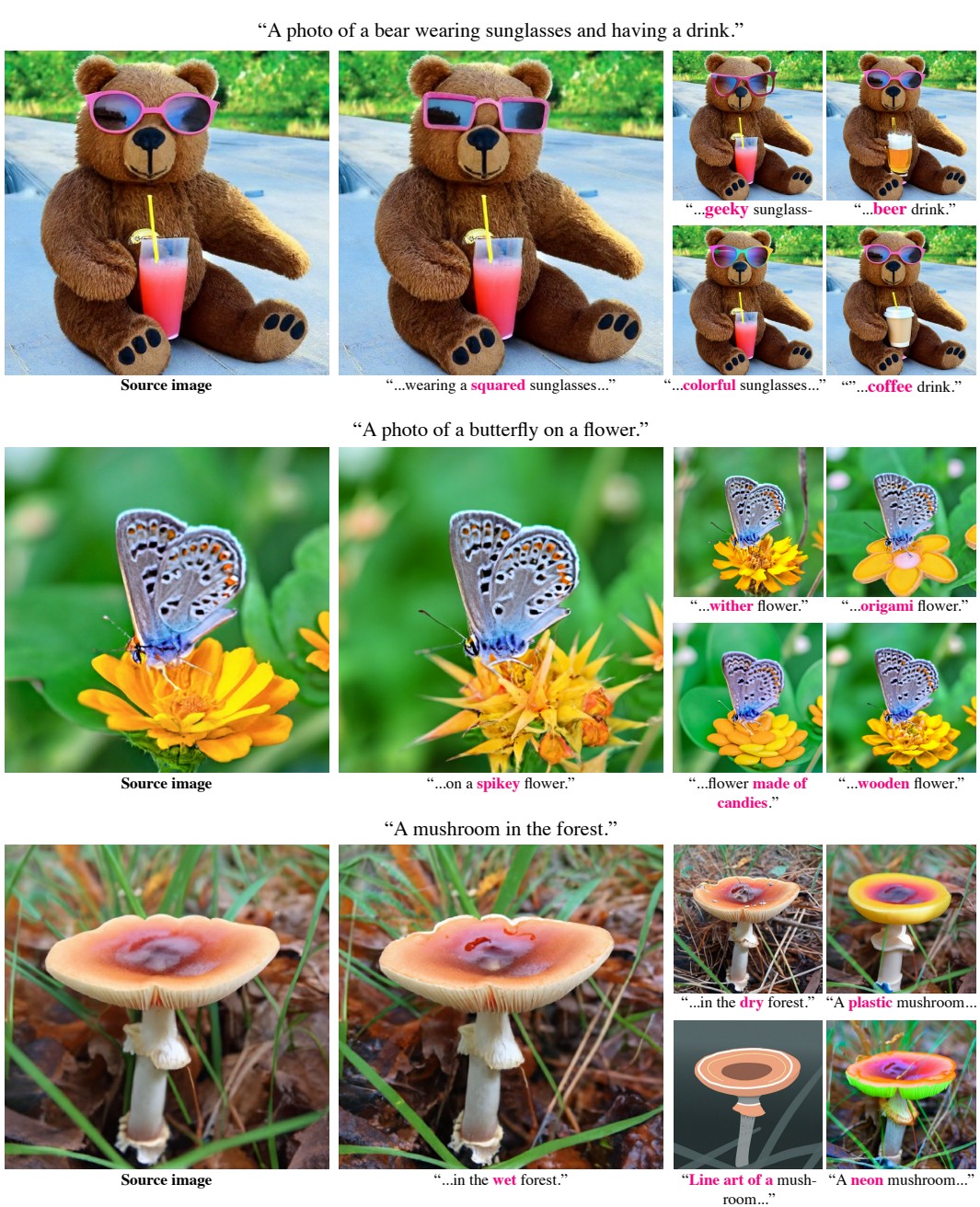

Figure 23: Additional results for Prompt-to-Prompt editing by adding a specification using the Imagen model (Saharia et al., 2022b)..

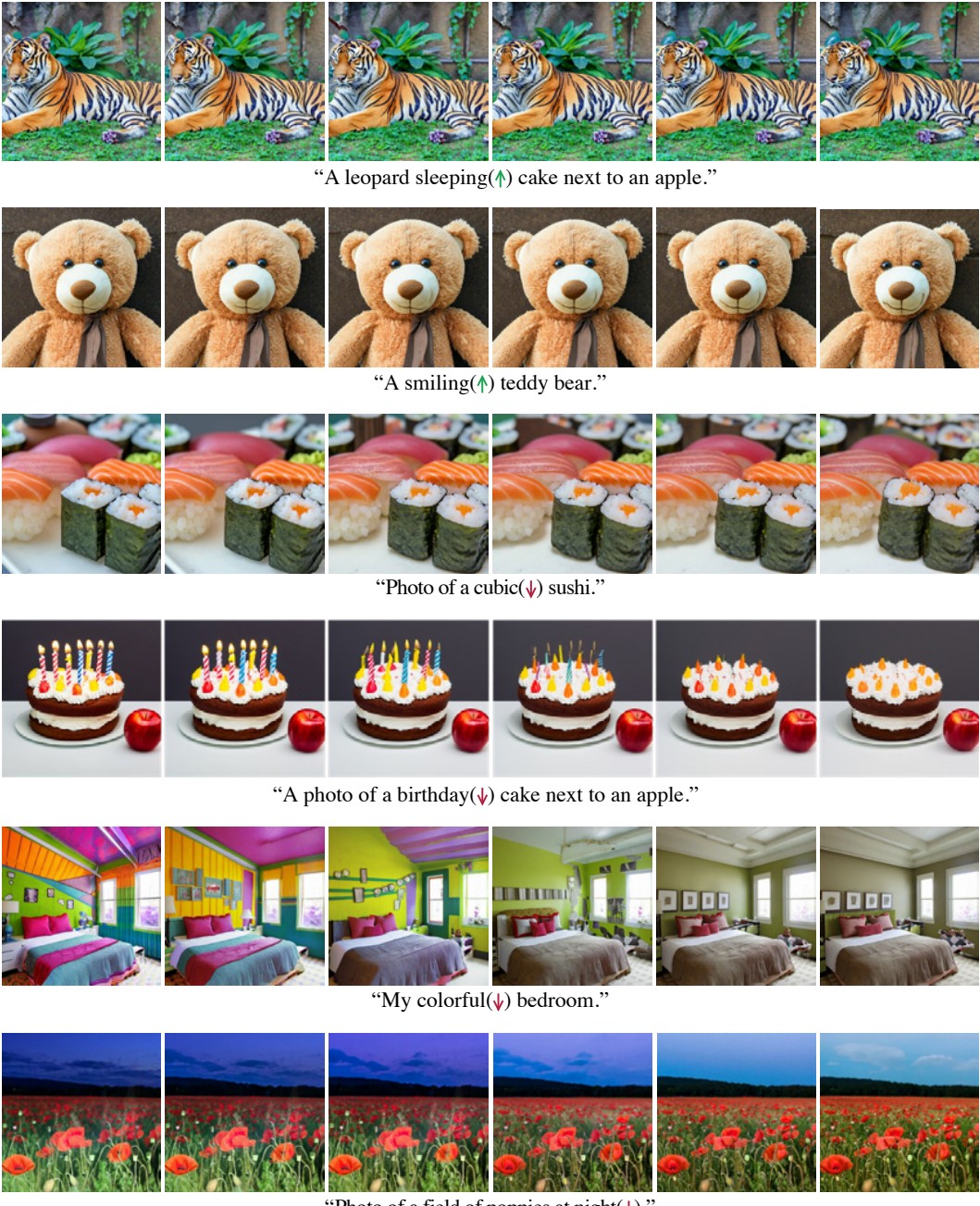

"A leopard sleeping(⬆) cake next to an apple."

"A smiling(⬆) teddy bear."

"Photo of a cubic(⬇) sushi."

"A photo of a birthday(⬇) cake next to an apple."

"My colorful(⬇) bedroom."

"Photo of a field of poppies at night(⬇)."

Figure 24: Additional results for Prompt-to-Prompt editing by attention re-weighting using the Imagen model (Saharia et al., 2022b).

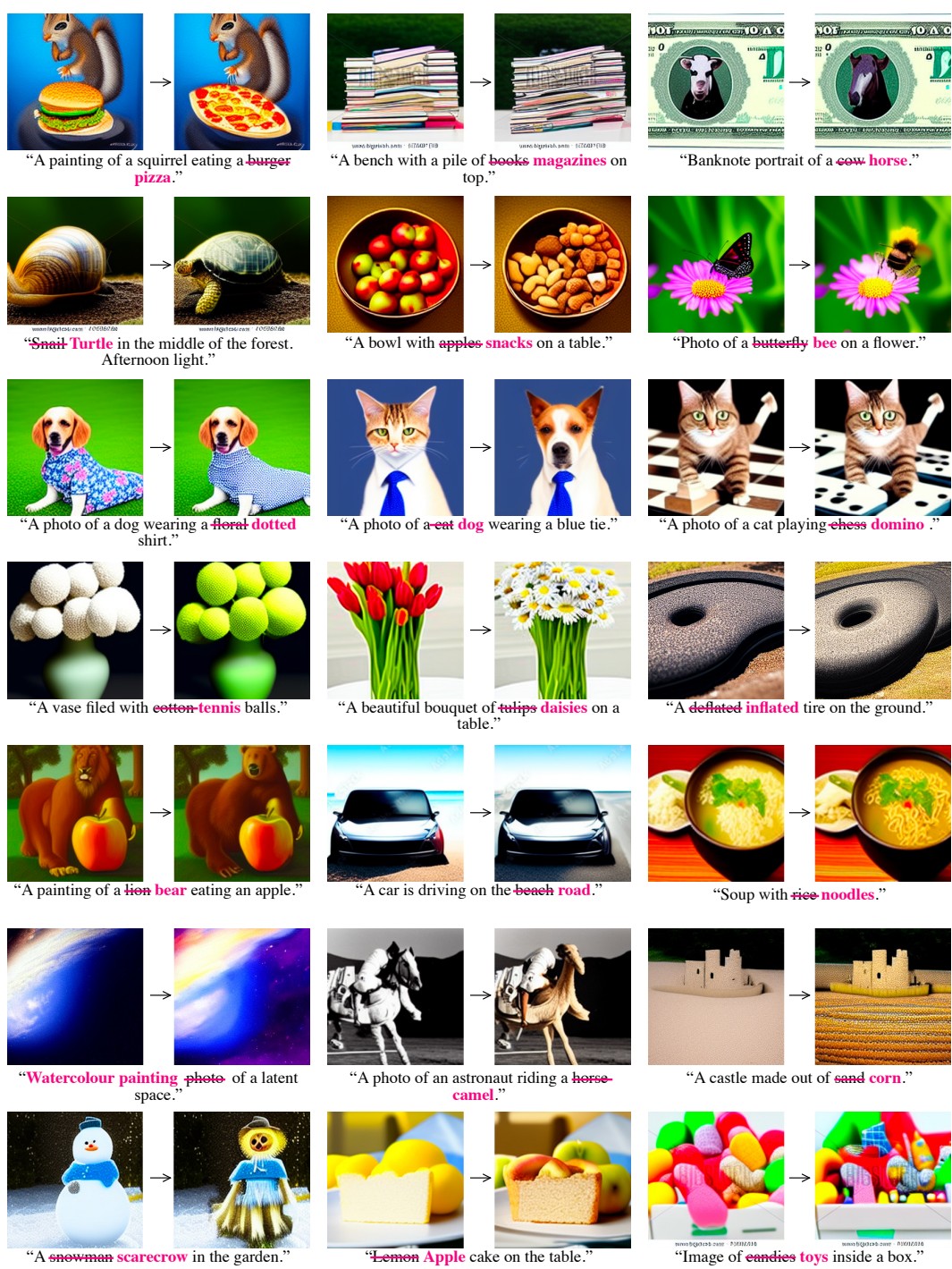

Figure 25: Additional results for Prompt-to-Prompt editing by word swap using the Latent Diffusion Model (Rombach et al., 2021).

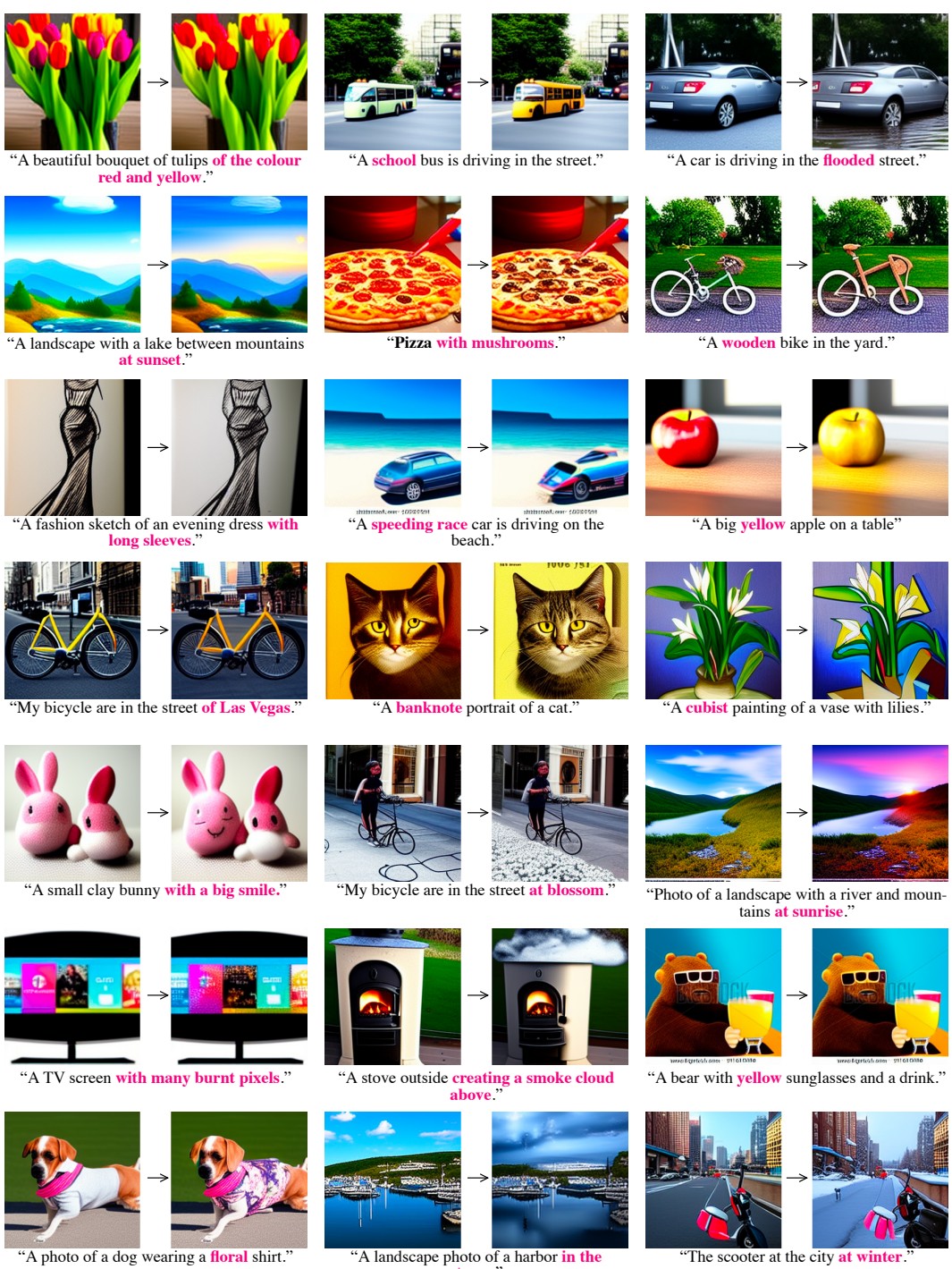

Figure 26: Additional results for Prompt-to-Prompt editing by adding a specification using the Latent Diffusion Model (Rombach et al., 2021).

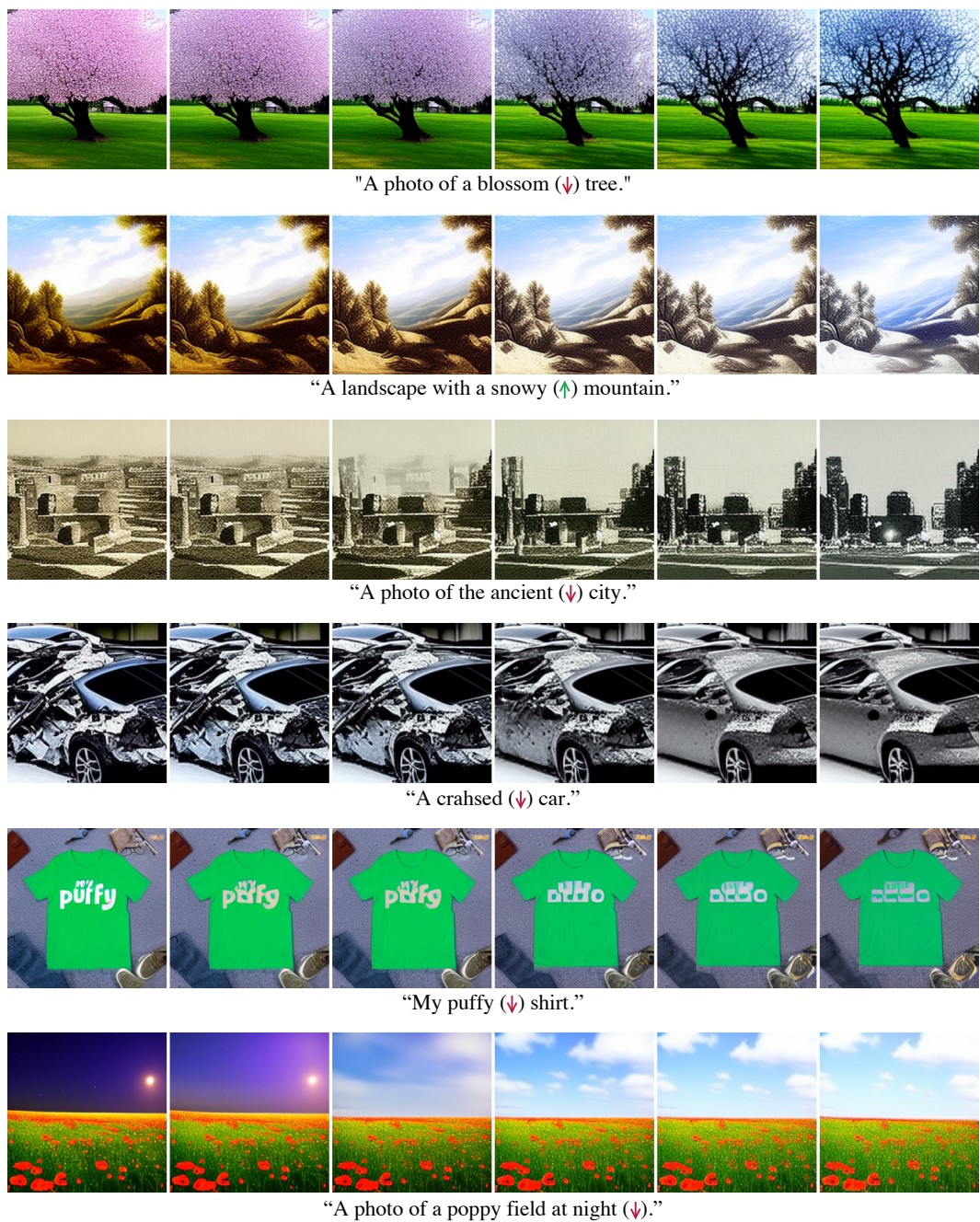

Figure 27: Additional results for Prompt-to-Prompt editing by attention re-weighting using the Latent Diffusion Model (Rombach et al., 2021).

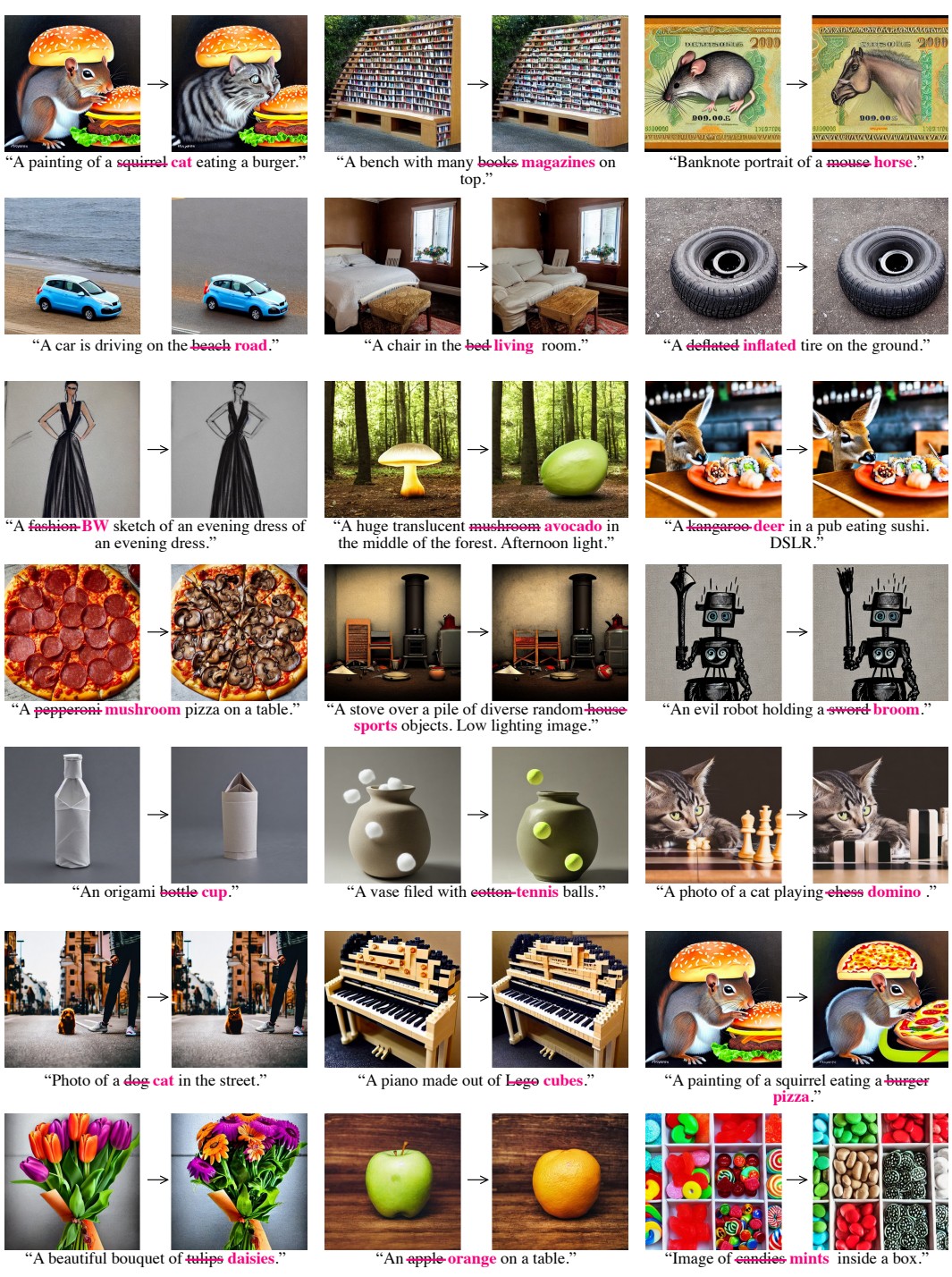

Figure 28: Additional results for Prompt-to-Prompt editing by word swap using the Stable Diffusion Model .

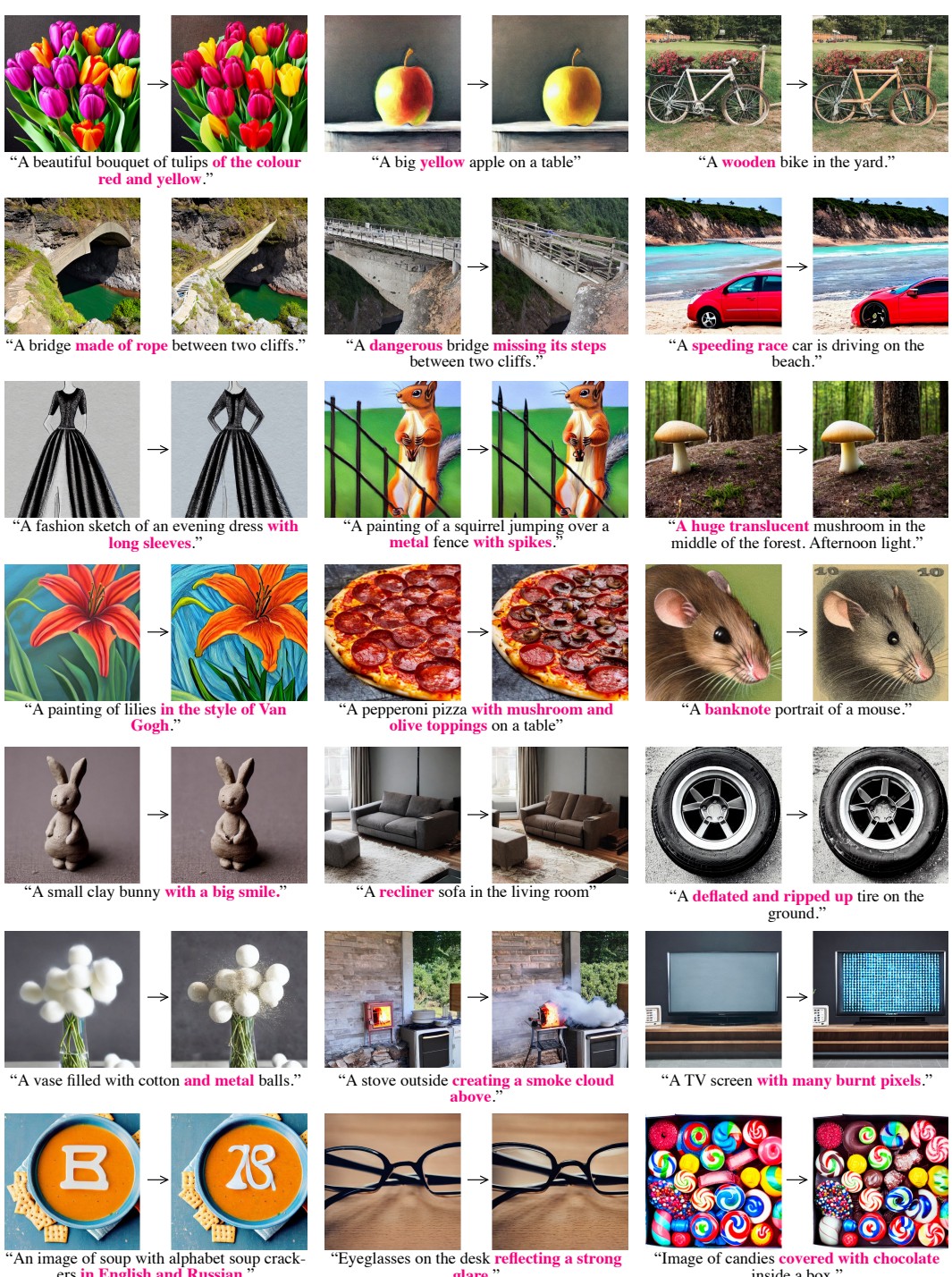

Figure 29: Additional results for Prompt-to-Prompt editing by adding a specification using the Stable Diffusion Model.

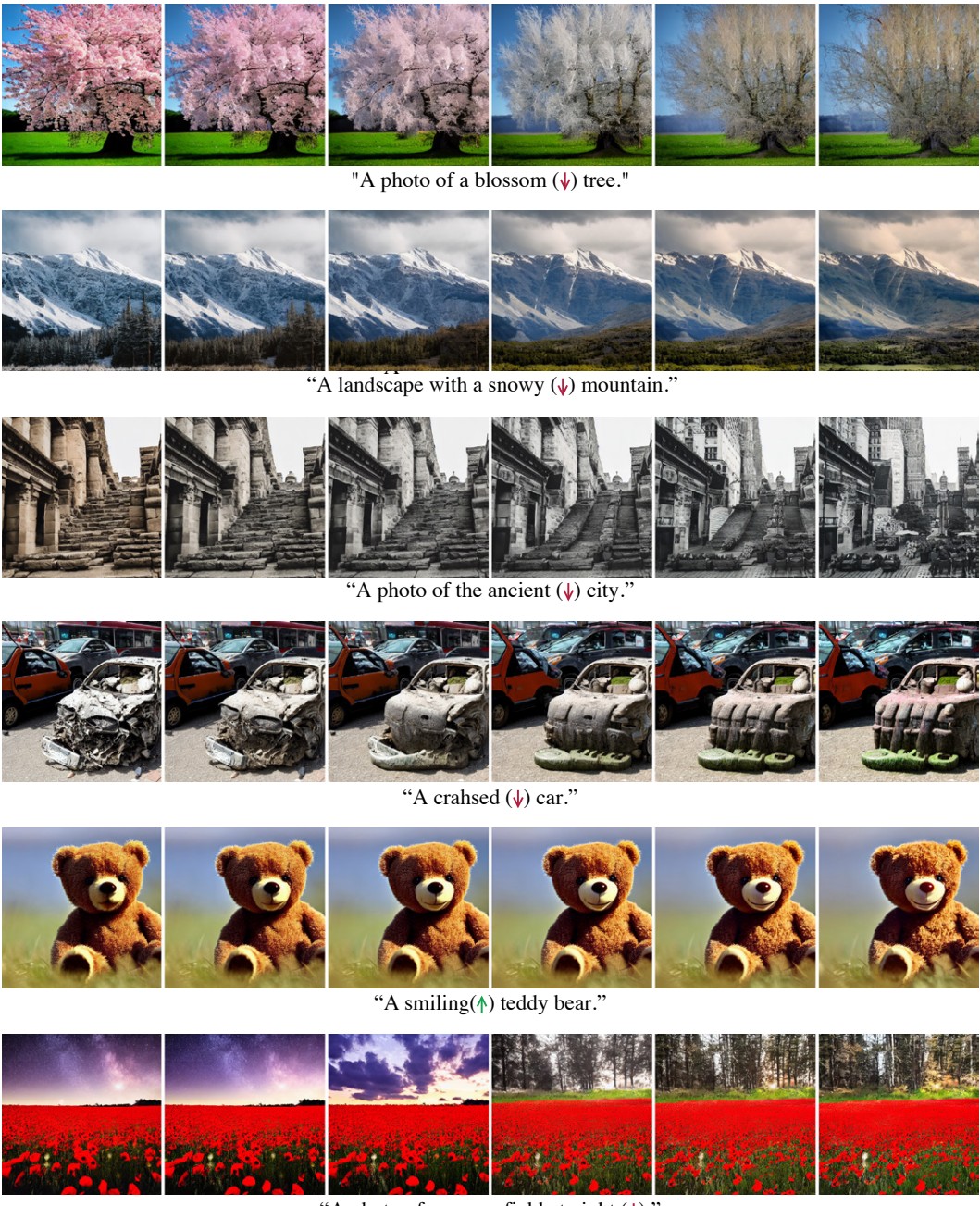

Figure 30: Additional results for Prompt-to-Prompt editing by attention re-weighting using the Stable Diffusion Model.

Figure 31: **Screenshots from our User study.** *The participants were asked to evaluate: (1) background, structure, and content preservation with respect to the source image, (2) alignment to the text, and (3) realism. The study evaluates both local and global editing.*

