# OpenReview forum: "Prompt-to-Prompt Image Editing with Cross-Attention Control"
_ICLR.cc/2023/Conference — ICLR 2023 notable top 25%_

### Official Review · Reviewer_5o2W · 2022-10-22

**Confidence:** 4
**Correctness:** 4
**Technical Novelty And Significance:** 3
**Empirical Novelty And Significance:** 4
**Recommendation:** 8

**Clarity, Quality, Novelty And Reproducibility:**

**Clarity:** The paper is clearly written and was easy to follow and read for me. I didn't spot any typos or other mistakes.

**Quality:** I think the overall quality of the work is high. State-of-the-art models are used. Appropriate experiments are run, including a user study. All evaluations seem correct. Appropriate baselines have been used in comparisons.

**Novelty:** The approach is novel, to the best of my knowledge. The finding that the attention maps in the cross attention layers capture the structure of the text-image alignment and can even be re-used in new synthesis processes is significant and could find re-use in other works.

**Reproducibility:** The reproducibility of the work is somewhat limited. While the work also uses the publicly available Latent Diffusion Model and Stable Diffusion, the main model used in most experiments is Imagen, which is not publicly available and cannot easily be re-trained.

**Strength And Weaknesses:**

**Strengths:**

- The proposed idea, modifying and re-using the attention maps of a diffusion model from one image, when generating a new one, is simple, yet elegant and novel. The approach makes intuitively sense and works well in practice.
- Generally, the paper is well-written, well-motivated and easy to follow (also see below). The work is also appropriately put into the broader context in the literature.
- Experimentally, the editing results are quite strong and the original structure and composition of the edited images is usually indeed well preserved when a new text prompt is used for editing. The fact that this is purely text-driven and yet allows fairly fine control is appealing.

**Weaknesses:**

- The work is only partially reproducible. The Imagen model is not available to the public and cannot be re-trained easily. The results on Stable Diffusion and the Latent Diffusion Model can be worse in some cases (see Appendix D.1).
- The approach comes with some hyperparameters. For instance, to achieve best results the attention map injection should only happen up to a certain time stamp during the iterative synthesis process. It seems like this time stamp would need to be tuned for each example separately, which can be costly, since running the diffusion model synthesis process repeatedly is expensive. This may not be ideal for interactive, real-time applications.
- Editing given real images is difficult, because this first requires an inversion process as well as finding an appropriate text prompt that could have generate the given image (this is discussed in the paper).
- As also pointed out by the authors themselves, other minor weaknesses include that the attention maps that are modified are in the low resolution layers of the U-Net, which prevents very fine-grained editing. Furthermore, editing corresponding to large-scale object movement is not possible with the proposed technique.

**Summary Of The Paper:**

The paper proposes a text-driven image editing technique to refine images generated from large-scale text-to-image diffusion models. The paper relies on a crucial observation: Just modifying a text prompt when calling the generator results in structurally and compositionally very different images, even if the text prompt is modified only slightly and even if the same random seed is used. However, we can use the attention maps of the cross attention layers (that attend to the text tokens) corresponding to the initial image, and inject them into the generation process when generating a new image with a modified prompt. Now the new image will correspond structurally and compositionally to the original one, while also reflecting the new text prompt. This allows the authors to demonstrate rich text-driven editing capabilities. The method can be combined with masking schemes to localize the edits and re-weighting techniques to give more or less weight to certain words in the prompt. The approach is compared to several baselines and shows favourable results.

**Summary Of The Review:**

In summary, I think this is a strong paper, which I recommend for acceptance. The methodology is based on a smart idea and novel, and the results are appealing. The paper is of high-quality throughout and well written and clearly presented. The paper has a couple of small weaknesses and the authors have mostly been transparent about that, but I do not think there are any major flaws. I believe the method could find practical use in the generative modeling community, for instance for producing generative art with even finer control.

---

> ### Author Response · Authors · 2022-11-07
> **R4 Response**
>
> We thank the reviewer for the valuable feedback.
>
> For reproducibility, please refer to the general response.
>
> As the reviewer kindly noticed, most mentioned weaknesses were already discussed in our paper. Therefore, we address the hyperparameter concern:
>
> Q: “It seems like this time stamp would need to be tuned for each example separately, which can be costly” \
> A: Our method does require setting the timestamp parameter for attention injection and the scale parameter for attention re-weighting. However, we observe that our method is not highly sensitive to these and using a constant timestamp of 0.5 for cross-attention and 0.2 for self-attention produces satisfying results in most cases. In particular, we have used these parameters in the user study to automatically generate our results. Therefore, in practice, the timestamp parameter can be easily tuned in ~1 minute, since single editing takes up to 20 seconds (with the Imagen model). We believe that future works would reduce the inference time of these models, so tuning the parameter would be quicker and more intuitive. Furthermore, setting the parameter is also a major advantage as it allows fine and intuitive control over the induced effect. For instance, the user can control the tradeoff between preserving the original details and hallucinating new ones.
> In particular, the result of attention re-weighting is heavily dependent on the original generation and the user preference, and therefore using a different parameter for different editing operations makes sense. We added this limitation to Section 5 in the revision (marked in red).
> Note that the effect of each hyperparameter is already demonstrated in the paper, see Figures 10, 11, and 23.

---

> > ### Comment · Reviewer_5o2W · 2022-11-22
> > **Thank you for reply**
> >
> > Thank you for the reply and discussion on the timestamp and scale parameters. I have no further questions and maintain my already positive opinion of the work.

---

### Official Review · Reviewer_vsUX · 2022-10-24

**Confidence:** 4
**Correctness:** 4
**Technical Novelty And Significance:** 3
**Empirical Novelty And Significance:** 3
**Recommendation:** 8

**Clarity, Quality, Novelty And Reproducibility:**

I think the proposed method is technically sound and the results are very promising. They addressed a practical issue and solved using novel and reasonable techniques. The authors also provided various interesting editing applications which help evaluate the method's value.

The paper writing is clear and figures are informative. The quality and clarity are satisfying.

**Details Of Ethics Concerns:**

No ethics concerns.

**Strength And Weaknesses:**

Strengths
1. The proposed task of only using text for local image editing is very interesting and of good practical values.
2. The authors proposed to use cross attention to connect word and image regions, which is novel and reasonable. Although attention maps have been widely used in other tasks, I think it is quite appropriate here.
3. Many interesting applications have been provided. All results are promising and clearly show this method's advantages.
4. The paper writing is clear and easy to follow.

Weaknesses
1. Maybe some ablation study would be better.
For example, does the proposed method affected by different noise sampler?

2. For those tasks that only local edits are needed, is it possible to only process local area and accelerate the process?

**Summary Of The Paper:**

This paper proposes a new method to only use text to achieve local image editing based on recent text-to-image models. More specifically, they propose to use cross-attention to connect each word and image regions and get rid of user-provided masks. Many interesting applications have been proposed and the results are very promising.

**Summary Of The Review:**

This paper proposes a new method to only use text to achieve local image editing based on recent text-to-image models.
The method is novel and technically sound. And the author provided various new applications which further prove its effectiveness.

I think this method can also inspire many interesting future work in this area.

---

> ### Author Response · Authors · 2022-11-07
> **R3 Response**
>
> We thank the reviewer for the valuable feedback.
>
>
> Q: “Does the proposed method affected by a different noise sampler?” \
> A: Manipulating the attention maps is not related to the noise sampler. Therefore, we have used the default sampler for each network: DDPM for Imagen, PLMS for Stable diffusion, and DDIM for latent diffusion. Using different sampler results with the same editing effect. In particular, we have tried to use the DDIM for Imagen but didn’t witness any significant changes from the DDPM sampler besides the natural difference in the quality of these samplers. Therefore, we conclude that our method can be applied with any sampler and there is no added value in showing such an experiment.
>
>
> Q: “For those tasks that only local edits are needed, is it possible to only process local area and accelerate the process?” \
> A: In our setting, we perform the editing using only text. Then we use the obtained attention maps for localization. Therefore, an initial pass with the full resolution is required.
> Yet, for the Imagen model, one can think of performing the super-resolution only over the locally edited part. But, this requires an architectural modification as the network should support different resolutions, and so we consider it as out of our scope.

---

### Official Review · Reviewer_kTNM · 2022-10-24

**Confidence:** 4
**Correctness:** 4
**Technical Novelty And Significance:** 4
**Empirical Novelty And Significance:** 3
**Recommendation:** 6

**Clarity, Quality, Novelty And Reproducibility:**

The paper is well-written. Editing the image by text alone while keeping the structure is very interesting and novel.

**Details Of Ethics Concerns:**

None.

**Strength And Weaknesses:**

Strength
1. The paper enables the user to control the generated image by text alone, by keeping the structure.
2. The proposed framework allows the user to control the extent.

Weakness
1. Failure cases are not shown for future work. There would be some failure cases or unsolved types of text edition. If the paper can discuss this aspect, the paper would be more appealing to readers.

**Summary Of The Paper:**

The paper proposes an intuitive prompt-to-prompt editing framework, where the edits are controlled by text only. While large-scale language-image (LLI) models are collecting attention to generate an image from text, they have limitation to control by only text. The paper  propose to control the attention maps of the edited image by injecting the attention maps of the original image along the diffusion process. The paper shows diverse results by by editing the textual prompt only.

**Summary Of The Review:**

The paper tries to solve an issue of controlling the image by text alone in image editing based on large-scale language-image models. By analyzing the cross-attention layers, the paper nicely shows a way to control the image by text alone. Diverse results show the effectiveness.

---

> ### Author Response · Authors · 2022-11-07
> **R2 Response**
>
> We thank the reviewer for the valuable feedback.
>
> We now address the raised question:
>
> Q: “Failure cases are not shown for future work. There would be some failure cases or unsolved types of text edition.“ \
> A: Failure cases are discussed in Section 5 (Limitations). This includes our inability to move objects across the image, our inability to perform more precise editing due to the resolution of the attention maps, and our struggle to edit real images.
> We will add a figure with examples for such editing failure cases.
> We believe that these limitations would be a great starting point for future work.
> In addition, we have added another limitation to the revision (marked in red), considering the setting of the timestamp parameter - see the response to R4 for more details.
> Please let us know if you think that one of our limitations requires additional discussion to what is already presented in Section 5 or specific additional results.

---

### Official Review · Reviewer_uigg · 2022-10-27

**Confidence:** 5
**Correctness:** 4
**Technical Novelty And Significance:** 4
**Empirical Novelty And Significance:** 4
**Recommendation:** 8

**Clarity, Quality, Novelty And Reproducibility:**

The clarity and quality of the presentation is very high. It was a real pleasure to read this work. As far as I know, this approach is novel and not only advances the processing capabilities of text-image diffusion models, but also opens up
a path to their interpretation.

Reproducibility:
Most of the experiments were performed (and presumably developed) on a closed-source system (Imagen) and are therefore not reproducible.
Although the results are certainly impressive, this is a clear drawback.
However, the work applies the method to publicly available systems and compares their performance, confirming that the approach works well for different types of methods.


**Strength And Weaknesses:**

This is a strong empirical paper with convincing experimental results for editing with text-to-image diffusion models.
The finding that most of the object structure is stored in the cross-attention maps is interesting and has practical implications.
Several methods are presented for obtaining finer-grained, local control when editing text.
Finally, the manuscript is clearly written, easy to understand, and provides very good visual explanations.

A few questions that I had while reading the paper:
How much control is lost by only applying the model on the 64x64 base model? Is this different in models using a diffusion model for upscaling or the decoder of an autoencoder? Does the "attention caching" lead to an improvement in sampling speed?




**Summary Of The Paper:**

In this paper, an approach to controllable image editing with large-scale text-to-image diffusion models is proposed.
More specifically, an approach is presented that first analyzes the cross-attention (and self-attention) layers in the U-net backbone of state-of-the-art text-image diffusion models. It is then shown that these layers capture most of the structural information about the synthesized images.
This opens up a number of modification possibilities by directly modifying these attention maps. The paper demonstrates these modification possibilities using, among others, content modification and local semantic editing.
This overcomes a problem that arises when using "vanilla" prompt-based modifications, where small changes in the prompt can lead to very different synthesis outcomes.




**Summary Of The Review:**

This is a very good empirical work with practical implications that greatly extends the applicability of modern text-to-image systems. The only major drawback is the fact that the method was mainly tested and evaluated on a closed-source system, which makes reproducibility difficult. However, since the method has also been validated on open systems, I am happy to recommend its acceptance for ICLR 2023.

---

> ### Author Response · Authors · 2022-11-07
> **R1 Response:**
>
> We thank the reviewer for the valuable feedback.
>
> For reproducibility, please refer to the general response.
> We now address the raised questions:
>
>
>
> Q: “How much control is lost by only applying the model on the 64x64 base model?”  \
> A: We observe that most details, including the composition, object structure, and colors are determined at a low resolution. These are difficult to control and thus we focus on these in our work.
> On the other hand, fine textures are determined at the super-resolution stage and are much easier to manipulate, since all other details were already determined in the 64X64 base model.
> Consider the following example - we generate a 64X64 image using the prompt “dog is sitting in the grass”. By simply feeding a different prompt *only* to the super-resolution we can manipulate the texture, e.g., “bushy grass” or “smooth grass”. However, feeding the super-resolution with “beach” instead of grass would fail as the color of the grass was already determined by the 64X64 image.
> We do observe that the attention re-weighting application can benefit from applying our method to the super-resolution in the case of amplifying or attenuating textures, such as amplifying the “fluffiness” of a doll. We add this observation to the revision in Section 3.2 (“Attention Re–weighting” paragraph), marked in red.
> Note that this is relevant only to the Imagen model. Stable Diffusion, on the other hand, consists of a single diffusion model.
>
>
>
> Q: “Is this different in models using a diffusion model for upscaling or the decoder of an autoencoder?” \
> A: Imagen, which uses a diffusion model for upscaling, and Stable diffusion, which uses decoder for upscaling, are different in many aspects. In particular, we hypothesize that using a diffusion model for super-resolution improves the realism achieved by the Imagen model.
> However, we find that the difference in upscaling technique is actually irrelevant to our method, as we manipulate the low-resolution representations by injecting the attention maps. Instead, as we already discuss in appendix D.1, we find that our method is deeply affected by the different text embedding used by the models.
>
>
>
> Q: “Does the "attention caching" lead to an improvement in sampling speed?” \
> A: When the cross-attention maps are injected, computing the corresponding cross-attention layers can be skipped, which reduces the overall computation. However, most layers, such as convolution and self-attention, cannot be skipped, so we expect only a marginal improvement. More specifically, self-attention is more computationally expensive since the number of pixels is larger than the number of textual tokens, while the injection of self-attention occurs only for a small portion of the timestamps (~20%).

---

### Author Response · Authors · 2022-11-07
**General Response**

We thank all the reviewers for their detailed reviews and constructive remarks.
We were excited to see that they found the results convincing (R1,R2,R3,R4), the paper well-written (R1,R2,R3,R4), and our method to be novel (R1,R2,R3,R4).


We would happily address any additional questions and concerns during the discussion period. Individual questions are answered separately to each reviewer.

We now turn to address the shared concern regarding reproducibility:
Indeed, we have used a non-public model for most of the experiments.
Therefore, we will publish an official implementation for the publicly available Stable and Latent Diffusion models. The code repository is ready to release and will be available as soon as anonymity breaks. The code has already been tested and we have verified that this will allow anyone to easily experiment with our method.

In section 4.1 (“Different Backbone” paragraph), we already demonstrate that our method works well with Latent and Stable Diffusion, as our analysis of the cross-attention maps also applied to these. In addition, further analysis of the difference between Imagen and Stable Diffusion is already provided in appendix D.1.

---

### Decision · Program_Chairs · 2023-01-20

**Decision:**

Accept: notable-top-25%

**Justification For Why Not Higher Score:**

This is a nice application paper. The presented idea is only applied to a subset of DL models for a specific application. The impact it creates will be limited to a small sub-field.

**Justification For Why Not Lower Score:**

It is clearly a good paper with good motivation, presentation, algorithm design, and results.

**Metareview: Summary, Strengths And Weaknesses:**


The paper proposes a prompt-editing method for text-to-image diffusion models that utilize cross-attention layers in their design. It studies the following prompt-editing setting. There are a source prompt and a target prompt where the target prompt is derived from the source prompt. Only one visual concept differs in these two prompts. The goal here is to keep most of the source prompt-driven text-to-image output the same and only modify the part of the image where the source and target prompt disagree. The observation made in the paper is that such a task can be achieved by manipulating the attention maps for the cross-attention layers. The paper gives a clear exposition of why this can be achieved by manipulating the attention maps. The presented algorithm is easy to follow. The results are impressive.

The paper receives 4 reviews. All of them consider the paper above the bar, citing various strengths such as the paper being well-motivated, the algorithm being well-designed and easy to follow, and the application being useful and interesting. The meta-reviewer agrees with the assessment and would like to recommend acceptance of the paper.

**Note From Pc:**

if the above contains the word "oral" or "spotlight" please see: "oral" presentation means -> notable-top-5% and "spotlight" means -> notable-top-25%. As stated in our emails, we are disassociating presentation type from AC recommendations